# Light-inducible protein degradation in *E. coli* with the LOVdeg tag

**Nathan Tague[1,2], Cristian Coriano-Ortiz[1,2], Michael B Sheets[1,2], Mary J Dunlop[1,2]\***

[1]Department of Biomedical Engineering, Boston University, Boston, United States;
[2]Biological Design Center, Boston University, Boston, United States

**Abstract** Molecular tools for optogenetic control allow for spatial and temporal regulation of cell behavior. In particular, light-controlled protein degradation is a valuable mechanism of regulation because it can be highly modular, used in tandem with other control mechanisms, and maintain functionality throughout growth phases. Here, we engineered LOVdeg, a tag that can be appended to a protein of interest for inducible degradation in *Escherichia coli* using blue light. We demonstrate the modularity of LOVdeg by using it to tag a range of proteins, including the LacI repressor, CRISPRa activator, and the AcrB efflux pump. Additionally, we demonstrate the utility of pairing the LOVdeg tag with existing optogenetic tools to enhance performance by developing a combined EL222 and LOVdeg system. Finally, we use the LOVdeg tag in a metabolic engineering application to demonstrate post-translational control of metabolism. Together, our results highlight the modularity and functionality of the LOVdeg tag system and introduce a powerful new tool for bacterial optogenetics.

**\*For correspondence:** mjdunlop@bu.edu

**Competing interest:** The authors declare that no competing interests exist.

**Preprint posted** 26 February 2023
**Sent for Review** 20 March 2023
**Reviewed preprint posted** 30 May 2023
**Reviewed preprint revised** 03 January 2024
**Version of Record published** 25 January 2024

## eLife assessment

This **valuable** study reports on a new tool that allows for light-controlled protein degradation in *Escherichia coli*. With the improved light-responsive protein tag, endogenous protein levels can be reduced several fold. The methodology is **convincing** and will be of interest to the fields of gene expression regulation in bacteria and more generally to synthetic biologists.

## Introduction

Currently, the most widely used optogenetic systems in bacterial synthetic biology rely on transcription to modulate gene expression. Light-responsive transcription factors allow genes of interest to be controlled modularly without having to re-engineer promoters. These tools have provided a valuable means to probe regulatory networks (*Dessauges et al., 2022*; *Harrigan et al., 2018*; *Lugagne and Dunlop, 2019*; *Olson et al., 2014*). However, these networks can be governed by complex control, where post-transcriptional and post-translational mechanisms work in concert with transcriptional regulation (*Chubukov et al., 2014*; *Link et al., 2013*; *Mahmoud and Chien, 2018*). Light-responsive regulators that act beyond transcription are necessary to more fully mimic natural biology and have the potential to improve upon current deficits of transcriptional control. In synthetic contexts, transcriptional regulation suffers from limited response dynamics. For example, in the case of gene deactivation, decreasing protein abundance is limited by cell division-based dilution or natural protein half-lives, which are in the range of 5–20 hr for the majority of proteins in *Escherichia coli* (*Maurizi, 1992*). When cells are growing in exponential phase, the effective degradation rate of these long-lived proteins is controlled by the cell cycle time, which is on the order of 0.5 hr. However, at high cell densities, such as during stationary phase, slow growth rates result in protein half-lives on the scale of tens of hours (*Maier et al., 2011*). Subpopulations of cells in stationary phase have been reported to

proliferate; however, these growing subpopulations are small compared to non-growing cells and add to heterogeneous protein half-lives within the population (*Jõers et al., 2020*), making it challenging to reliably remove proteins. It should be noted that a proportion of proteins are actively degraded during slow growth, but these represent the minority in *E. coli* (*Gupta et al., 2022*). This is a critical issue for metabolic engineering where chemical production is typically carried out at stationary phase in two-stage fermentations (*Lalwani et al., 2018*).

Post-translational optogenetic control has the potential to address some of the shortcomings of transcriptional regulation because post-translational control mechanisms can function independently of growth-based dilution. Existing approaches to post-translational optogenetic control include the use of light-inducible dimers in split protein systems (*Baumschlager et al., 2017*; *Kawano et al., 2015*; *Nihongaki et al., 2015*; *Tague et al., 2023*), domain insertions with light-controlled allosteric domains (*Dagliyan et al., 2016*; *Gil et al., 2020*; *Zhu et al., 2023*), and membrane-confined functional control (*Strickland et al., 2012*; *Wang et al., 2016*). These approaches, however, suffer from a lack of modularity, which is a key benefit of the transcriptional control approaches. Split protein systems and domain insertions can require significant engineering for each protein of interest and solutions for a given protein are unlikely to map to proteins with different structures. Likewise, localization-based approaches are only applicable to a small subset of proteins that have position-dependent function, such as transcription factors that require nuclear localization (*Yumerefendi et al., 2018*).

Protein degradation offers potential as a modular post-translational mechanism of control. Endogenous proteolytic machinery is necessary for proteome homeostasis and acts as a global regulatory system (*Mahmoud and Chien, 2018*). Targeted proteolysis has proven useful in many synthetic biology applications in both eukaryotic and prokaryotic systems (*Andersen et al., 1998*; *Cameron and Collins, 2014*; *McGinness et al., 2006*; *Morreale et al., 2022*; *Trauth et al., 2019*). In eukaryotic cells, light-dependent protein degradation has been demonstrated using various mechanisms (*Bonger et al., 2014*; *Deng et al., 2020*; *Liu et al., 2020*; *Renicke et al., 2013*; *Xue et al., 2019*). For example, Bonger et al. utilized the LOV2 domain of *Avena sativa* phototropin 1 (*As*LOV2) to achieve light-dependent protein degradation in mammalian cells (*Bonger et al., 2014*). LOV2 is a blue-light-responsive protein that is widely used in optogenetic tools (*Pudasaini et al., 2015*). LOV2 contains a core Per-Arnt-Sim (PAS) domain surrounded by N- and C-terminal α-helices. Upon blue light illumination, the LOV2 protein undergoes a conformational change where the C-terminal Jα helix reversibly unfolds and becomes unstructured (*Halavaty and Moffat, 2007*; *Harper et al., 2003*; *Yamamoto et al., 2009*). Incorporation of a degradation-targeted peptide sequence into the C-terminal results in light-dependent protein degradation in yeast (*Renicke et al., 2013*).

However, the bacterial proteasome differs from the eukaryotic proteasome in many ways (*Mahmoud and Chien, 2018*; *Schrader et al., 2009*). The bacterial proteasome includes several proteases with divergent targeting behaviors and does not utilize ubiquitin as a generalized modification to trigger degradation (*Finley, 2009*). In bacteria, protein targeting for degradation is predominantly dependent on primary amino acid sequence as opposed to a ubiquitin-like appendage and, apart from certain well-studied cases, the rules governing sequence recognition of bacterial degrons are not fully understood (*Baker and Sauer, 2006*; *Striebel et al., 2009*). Because of these complexities, light-dependent degradation systems for bacteria have lagged behind their eukaryotic counterparts, where interaction of a ubiquitin ligase with a defined degron can be used as a control mechanism (*Bondeson et al., 2022*). Nevertheless, optogenetic degradation remains a key target for bacterial synthetic biology applications. In principle, an optogenetic degradation system could be fast-acting, modular, and interface with endogenous proteasome machinery. These features would provide a straightforward way of adding dynamic control to a protein of interest, complementing existing transcriptional tools for optogenetic control of gene expression. One recent system developed by Komera et al. utilizes a light-responsive split TEV protease to expose or remove constitutively active degradation tags in *E. coli* (*Komera et al., 2022*). Although this system achieves protein degradation in response to light, it acts indirectly through activation of an exogenous protease. A simpler approach where the degradation tag itself is light responsive would streamline this by eliminating the need for multiple exogenous components.

Here, we develop LOVdeg, a modular protein tag based on the *As*LOV2 protein that is conditionally degraded in response to blue light in *E. coli*. We demonstrate that attaching this tag to a protein of interest confers light-dependent protein instability. We show the modularity of the LOVdeg tag by

incorporating it into multiple proteins with widely varying function, converting them all into optogenetically controlled systems. In addition, through photocycle-stabilizing mutations, we create a version of the LOVdeg tag that responds to infrequent exposure to blue light. We also demonstrate that our degradation tag can be used in concert with other optogenetic systems for multilayer control by using EL222, a blue-light-responsive system for transcriptional control, together with LOVdeg. Lastly, we incorporate optogenetic degradation into a metabolically engineered strain to control production of octanoic acid. Overall, this work introduces a new bacterial optogenetic tool that overcomes several drawbacks of transcriptional control by providing post-translational degradation, while avoiding the need for substantial protein engineering that can be required for alternative post-translational control mechanisms.

## Results

### Design and characterization of the *As*LOV2-based degradation tag

The *E. coli* proteasome consists of five AAA+ proteases and is continuously active, either degrading misfolded proteins for quality control or balancing regulatory protein levels (*Mahmoud and Chien, 2018*). We set out to exploit the endogenous proteasome activity in order to design a light-responsive protein tag. To do this, we took insight from studies related to native protein quality control. In bacteria, peptides from stalled ribosomes are targeted for degradation through interaction with a tmRNA that appends a short amino acid sequence, known as an SsrA tag, to the C-terminal end of the incomplete protein (*Keiler, 2015*). The *E. coli* SsrA tag has been studied extensively and is known to interact with the unfoldases ClpX and ClpA (*Flynn et al., 2001*; *Gottesman et al., 1998*). Addition of the SsrA peptide sequence to exogenous proteins targets them for degradation by the host proteasome. SsrA-mediated degradation has proven useful in synthetic gene circuit function and biochemical production (*Elowitz, 2000*; *Gurbatri et al., 2020*; *Stricker et al., 2008*; *Torella et al., 2013*; *Ye et al., 2021*). A recent structural study of ClpX interacting with the SsrA tag from *Fei et al., 2020* demonstrated that in order for ClpX to unfold a tagged protein, the C-terminal tail needs to be unstructured and sufficiently long to fit into the ClpX pore. We reasoned that the mechanism of *As*LOV2, in which the C-terminal Jα helix becomes unstructured upon blue light absorption, could be utilized to provide light-inducible protein degradation.

Biochemical studies have probed the amino acid sequence of the SsrA tag and its role in degradation targeting. *Flynn et al., 2001* demonstrated that ClpA and ClpX unfoldases interact with overlapping residues within the SsrA sequence and that the last three amino acids (L-A-A) are particularly important for successful degradation. Importantly, they also showed that mutation of the leucine in the C-terminal 'L-A-A' lowers unfoldase affinity but does not hinder degradation completely. We noticed that the C-terminal amino acid sequence of the *As*LOV2 domain, comprised of residues 404–546 of *Avena sativa* phototropin 1, are 'E-A-A' at positions 541–543 (*Figure 1a*). The dark state structure of the native *As*LOV2 domain (PDB: 2V1A) shows that these three amino acids and K544 complete the folded Jα helix (*Figure 1b*). A truncation of residues 544–546 leaves the Jα helix largely intact and the resulting C-terminal 'E-A-A' remains caged as part of the folded helix as seen upon examination of the dark state structure (*Figure 1b*). We hypothesized that this truncation would be stable in the dark state as a consequence of 'E-A-A' caging and unstable in the light state due to Jα helix unfolding and exposure of an unstructured degradation tag.

Optogenetic systems have used the *As*LOV2 domain in *E. coli*; however, it is often appended N-terminally or internally to another protein (*Li et al., 2022*; *Strickland et al., 2008*). To the best of our knowledge, wild-type or truncated *As*LOV2, with its C-terminal end exposed, have not been used for proteolytic degradation in bacterial cells. To test the stability of C-terminal *As*LOV2 in *E. coli* in response to light, we constructed a plasmid where we used an IPTG-inducible promoter to control the expression of mCherry translationally fused to *As*LOV2 (*Figure 1c*). In this construct, the full C-terminal sequence is intact so unfoldases are not expected to have good access for protein degradation. Consistent with this, induction of mCherry-*As*LOV2(546) with IPTG increased expression of the fusion construct, and 465 nm blue light induction resulted in only a modest decrease in expression (*Figure 1d*). Next, we tested a version of the construct where three amino acids were truncated to expose the C-terminal 'E-A-A', *As*LOV2(543). This construct destabilized the protein fusion as predicted, resulting in significantly lower protein expression compared to the non-truncated version

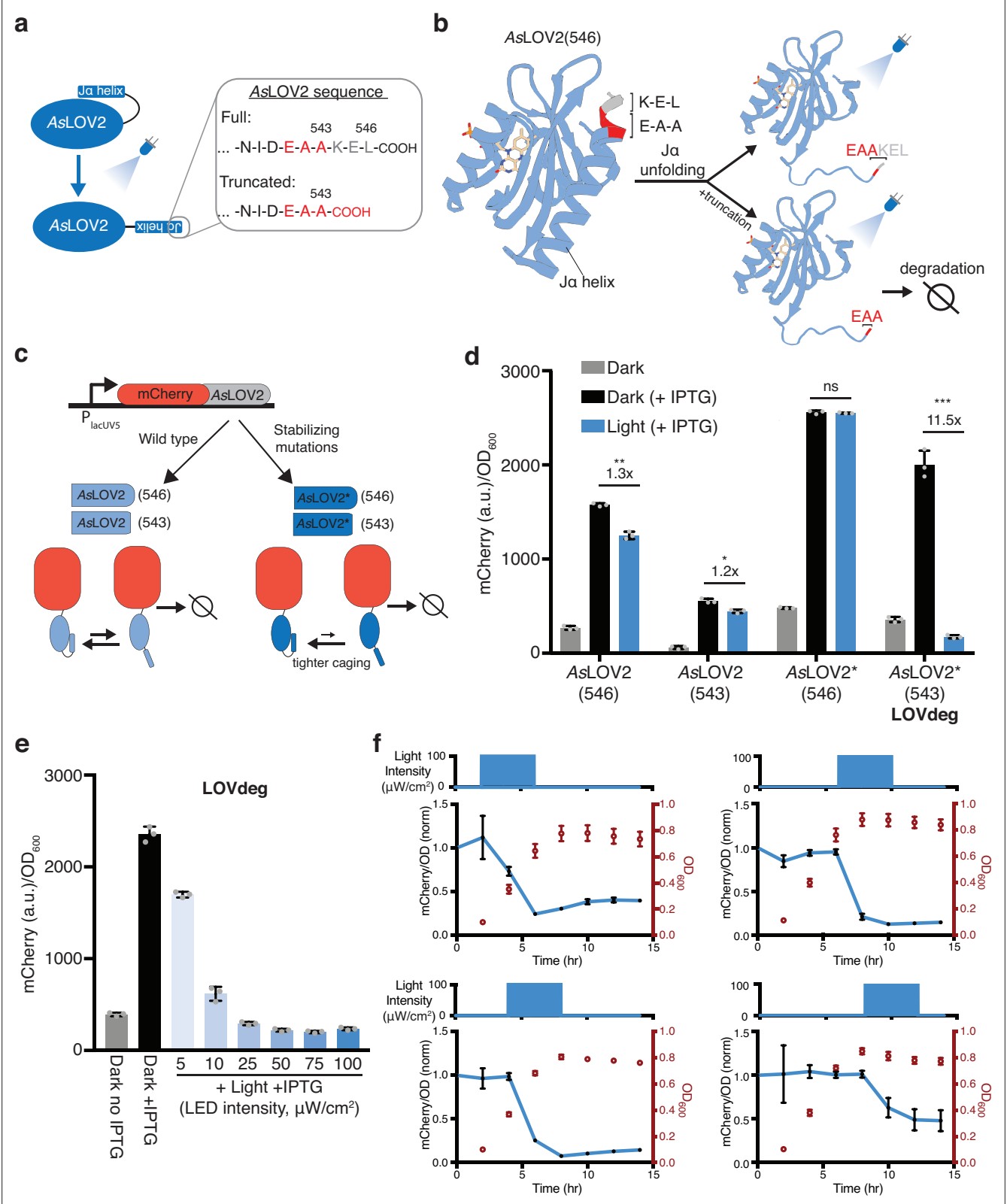

**Figure 1.** Design of *As*LOV2-based degradation tag. (**a**) Primary sequence of *As*LOV2(546) C-terminal sequence. A three amino acid truncation exposes E-A-A. (**b**) Structure of *As*LOV2 (aa404-546, PDB: 2V1A). Amino acids 541–543 (E-A-A) are red and 544–546 (K-E-L) are gray at the C-terminal of the Jα helix. (**c**) Construct used to characterize optogenetic control using *As*LOV2 variants. Each variant is translationally fused to mCherry expressed from an IPTG-inducible promoter. Variants include wild-type *As*LOV2 (light blue) and a dark state-stabilized version, AsLOV2* (dark blue), with and without

*Figure 1 continued on next page*

*Figure 1 continued*

the three amino acid truncation. (**d**) mCherry protein levels in response to 465 nm blue light for wild-type *AsLOV2*, and mutated *AsLOV2*\* fusions with and without truncation. *AsLOV2*\*(543) is the variant we denote the 'LOVdeg' tag. (\*\*\*p<0.0001; \*\*p<0.001; \*p<0.01; n.s., not significant; two-tailed unpaired *t*-test; n = 3 biological replicates). (**e**) mCherry-LOVdeg in response to variable light intensities. (**f**) mCherry fluorescence levels and optical density of mCherry-LOVdeg with 4 hr of 465 nm blue light exposure applied at different points in the growth cycle. Light exposure programs are plotted above each subplot and are staggered 2 hr apart (starting at 2, 4, 6, or 8 hr), all lasting 4 hr. Expression levels are normalized to the dark state control (*Figure 1—figure supplement 8*). Error bars show standard deviation around the mean.

The online version of this article includes the following figure supplement(s) for figure 1:

**Figure supplement 1.** Alignment of *AsLOV2*, iLID (mutated version of *AsLOV2* from *Guntas et al., 2015*) and LOVdeg (*AsLOV2*\*(543)) amino acid sequences.

**Figure supplement 2.** mCherry expression levels without IPTG induction in response to 465 nm blue light for wild-type *AsLOV2* and mutated *AsLOV2*\* fusions with and without truncation.

**Figure supplement 3.** Investigating proteasome components involved in LOVdeg tag destabilization.

**Figure supplement 4.** Expression of mCherry-LOVdeg over time under light exposure in wild-type cells, cells expressing exogenous HslUV, and cells expressing exogenous ClpA.

**Figure supplement 5.** Untagged mCherry expression induced with IPTG in wild-type and *clpX* knockout strains.

**Figure supplement 6.** Light-dependent stability of mCherry fusions with truncated and non-truncated LOVdeg tags in strains lacking *clpP* and *clpS*.

**Figure supplement 7.** Light response of mCherry without any *AsLOV2* variant.

**Figure supplement 8.** mCherry-LOVdeg fluorescence levels and growth without any light exposure.

**Figure supplement 9.** Comparing LOVdeg and iLID-SsrA.

**Figure supplement 10.** Phase contrast and fluorescence images of cells constitutively expressing mCherry-LOVdeg exposed to blue light or kept in the dark (scale bar = 10 μm).

(*Figure 1d*). However, counter to our expectations, we observed only a modest decrease in protein levels upon blue light induction.

While *AsLOV2* switches from mostly folded in the dark state to mostly unfolded in the light state, both states are present at an equilibrium with or without light (*Yao et al., 2008*). Due to the suboptimal equilibrium of native *AsLOV2*, dark state undocking and unfolding of the Jα helix is a common issue in *AsLOV2*-based optogenetic tools, and many studies have aimed at improving the dynamic range of *AsLOV2* (*Guntas et al., 2015*; *Lungu et al., 2012*; *Strickland et al., 2012*; *Strickland et al., 2010*; *Wang et al., 2016*). Guntas et al. performed phage display on an *AsLOV2*-based protein that incorporates a caged peptide sequence in the Jα helix. They identified a variant with 11 amino acid substitutions with much tighter dark state caging, called iLID. Interestingly, several of these mutations do not have a direct interaction with the caged peptide, but instead are located at the hinge loop connecting the PAS domain to the Jα helix or in the PAS domain itself (*Figure 1—figure supplement 1*). Guntas et al. further characterized amino acid substitutions and determined several that did not affect caging or dynamic range if reverted (F502Q, H521R, and C530M). We focused on the mutations found to result in tighter dark state caging (L493V, H519R, V520L, D522G, G528A, E537F, N538Q, and D540A) and reasoned that this set should stabilize the Jα helix irrespective of the caged peptide. Thus, we next constructed a variant of *AsLOV2* with these eight mutations, which we denote as *AsLOV2*\*.

To test whether these mutations are beneficial in the context of protein stabilization, we fused *AsLOV2*\*(546) or *AsLOV2*\*(543) to mCherry (*Figure 1c*). mCherry with non-truncated *AsLOV2*\*(546) expressed highly with the addition of IPTG, but blue light did not result in a decrease in mCherry levels, consistent with lack of access by proteases (*Figure 1d*). In contrast, the truncated version containing the mutations, *AsLOV2*\*(543), displayed the desired light-induced degradation, expressing highly in the dark and exhibiting 11.5× lower expression in response to blue light (*Figure 1d*). We also confirmed that *AsLOV2*\*(543) expression was decreased in response to light without any IPTG induction (*Figure 1—figure supplement 2*). Interestingly, *AsLOV2*\*(543) is degraded more in response to light relative to *AsLOV2*(543). We conducted experiments to identify the mechanism underlying *AsLOV2*\*(543) degradation and found evidence that ClpA is involved, and other proteases and unfoldases play complementary roles (Supplementary text, *Figure 1—figure supplements 3–6*). We also verified that light exposure itself did not affect mCherry protein levels or growth by conducting experiments with an IPTG-inducible mCherry without any *AsLOV2* variant fused and exposing cells to blue

light (*Figure 1—figure supplement 7*). We chose to move forward with the *As*LOV2*(543) variant due to its light-inducible properties, and we denoted this variant as the 'LOVdeg' tag.

Next, we used the LOVdeg tag to test the impact of light intensity and the timing of light exposure. We found that the LOVdeg tag is destabilized in proportion to light intensity (*Figure 1e*), making it straightforward to tune degradation by adjusting light levels. In addition, a key advantage of protein degradation-based mechanisms is that they should function at a range of growth phases. We tested this by varying when we applied blue light, ranging from early exponential to stationary phase. We found that light-dependent decreases in protein levels can be achieved at various stages of growth (*Figure 1f*, *Figure 1—figure supplement 8*).

To benchmark the degradation capacity of the LOVdeg tag, we compared it to protein levels of mCherry subject to constitutive degradation via addition of an SsrA tag. In order to maintain comparable transcription and translation, we appended an iLID *As*LOV2 derivative, which was modified to contain a full-length SsrA sequence, to the IPTG-inducible mCherry. As expected, mCherry-iLID-SsrA expressed poorly independent of IPTG induction (*Figure 1—figure supplement 9*). To quantify the light-dependent degradation of the LOVdeg tag, we compared its dark state expression level to that of mCherry-iLID-SsrA. Using this comparison, light-induced degradation of the LOVdeg tag reached 6× degradation compared to 14× for the SsrA tag. However, the comparison between these two is convoluted by the time required to degrade the protein in response to light, making it challenging to directly compare these numbers.

A potential concern is that light-induced disorder of the Jα helix could result in a decrease in solubility and aggregation of the LOVdeg tag. To rule this out as the cause of the fluorescence decrease, we captured microscopy images of cells in dark and light conditions. The imaging confirmed a light-dependent decrease in mCherry expression without the formation of visible protein aggregates (*Figure 1—figure supplement 10*). Thus, the LOVdeg tag variant provides blue light-dependent protein degradation.

## Modularity of the LOVdeg tag

Post-translational control of protein function can require significant protein engineering for each use case (*Sheets et al., 2020*; *Tague et al., 2023*; *Zhu et al., 2023*). Degradation tags, by contrast, offer post-translational control that theoretically requires little to no protein engineering and is protein agnostic. To test the modularity of the LOVdeg tag, we incorporated optogenetic control into three systems with highly diverse functions and relevance to synthetic biology and biotechnology applications: the LacI repressor, CRISPRa activation, and the AcrB efflux pump.

First, we sought to test whether the LOVdeg tag could be fused to transcription factors to enable light-dependent regulation. The LacI repressor is a widely used chemically inducible system in synthetic biology. We translationally fused the LOVdeg tag to LacI and paired it with a reporter where the LacUV5 promoter controls expression of mCherry (*Figure 2a*). Our results show that light exposure successfully increased mCherry expression (*Figure 2b*). Light-induced mCherry expression did not achieve the full levels provided with saturating IPTG induction; however, we still observed a notable increase. We tested an alternative strategy for further improving de-repression, which suggested that the discrepancy between IPTG versus light-dependent induction likely stems from the delay in LacI degradation compared to the rapid allosteric action of IPTG (Supplementary text, *Figure 2—figure supplement 1*).

Next, we incorporated the LOVdeg tag into the SoxS-based bacterial CRISPRa activation system (*Dong et al., 2018*). In this system, a scaffold RNA, which is a modified gRNA containing an MS2 stem loop, is used to localize dCas9 and the transcriptional activator SoxS, which is fused to an MS2 coat protein (MCP). We translationally fused the LOVdeg tag to the MCP-SoxS protein, such that in the dark CRISPRa will be active and light exposure relieves activation (*Figure 2c*). In the original system, MCP-SoxS expression is anhydrotetracycline (aTc) inducible. This induction system in not amenable to blue light stimulation because aTc is photosensitive (*Baumschlager et al., 2020*). Thus, we changed the MCP-SoxS construct to an IPTG-inducible promoter prior to blue light experiments (*Figure 2—figure supplement 2a*, *Supplementary files 1 and 2*). We confirmed that the promoter switch maintained CRISPRa activity (*Figure 2—figure supplement 2b*). Fusing the LOVdeg tag to the activator component indeed relieved CRISPRa activity, resulting in a decrease in expression under blue light stimulation (*Figure 2d*). However, reversal of activator activity was not complete and expression during light

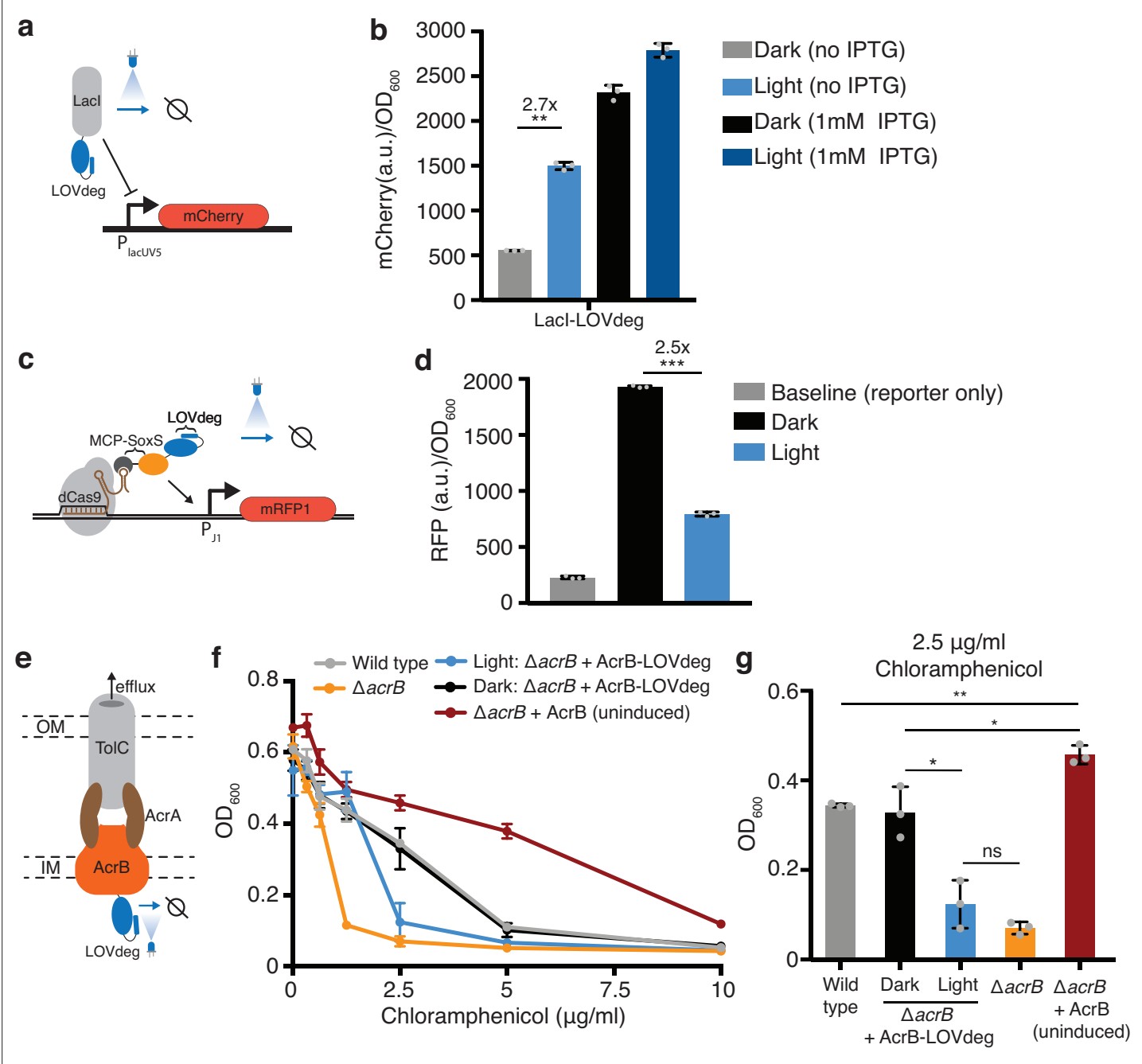

**Figure 2.** Incorporating light responsiveness into diverse proteins with the LOVdeg tag. (**a**) Control of mCherry repression using a LacI-LOVdeg fusion. (**b**) mCherry expression in response to light exposure for strains with LacI-LOVdeg compared to IPTG induction (**\*\*p<0.001, two-tailed unpaired *t*-test). (**c**) Schematic of SoxS-based CRISPRa activation with a LOVdeg tag appended to the MCP-SoxS activator. (**d**) CRISPRa control of mRFP1 expression in response to light (**\*\*\*p<0.0001, two-tailed unpaired *t*-test). (**e**) Schematic of the LOVdeg tag appended to AcrB of the AcrAB-TolC efflux pump. IM, inner membrane; OM, outer membrane. (**f**) Chloramphenicol sensitivity tests. Wild-type cells (BW25113) are compared to a Δ*acrB* (BW25113 Δ*acrB*) strain, Δ*acrB* complemented with AcrB-LOVdeg (Δ*acrB* + AcrB-LOVdeg) exposed to light or kept in the dark, and Δ*acrB* strain complemented with an IPTG-inducible AcrB (Δ*acrB* + AcrB). No IPTG was added to Δ*acrB* + AcrB or Δ*acrB* + AcrB-LOVdeg. (**g**) OD₆₀₀ of strains shown in (**f**) at 2.5 µg/mL chloramphenicol (**\*\*p<0.001; \*p<0.05; ns, not significant; two-tailed unpaired *t*-test). Error bars show standard deviation around the mean (n = 3 biological replicates).

The online version of this article includes the following figure supplement(s) for figure 2:

**Figure supplement 1.** LacI-LOVdeg and LacI-LOVdeg+decoy control of mCherry with 1 mM IPTG induction included.

**Figure supplement 2.** Switching promoters for the MCP-SoxS construct.

stimulation remained above the baseline levels from the reporter-only control, which could be due to the presence of low levels of the MCP-SoxS activator.

We also added the LOVdeg tag to the endogenous membrane protein AcrB. This represents a challenging test case for degrading a native protein. The AcrAB-TolC complex is a multidrug efflux pump with clinical relevance due to its role in antibiotic tolerance and resistance acquisition (*El Meouche and Dunlop, 2018*; *Lizarralde-Guerrero and Taraveau, 2021*; *Okusu et al., 1996*). AcrAB-TolC has also been utilized in metabolic engineering as a mechanism to pump out toxic chemical products and boost strain performance (*Dunlop et al., 2011*; *Fisher et al., 2014*). However, inducible control of AcrB is challenging because cell viability is sensitive to overexpression (*Turner and Dunlop, 2015*). Optogenetic transcriptional control systems have high dynamic ranges but do not operate in the low expression ranges relevant to very potent protein complexes such as AcrAB-TolC. Light-based degradation is well suited for this challenge because it works by decreasing protein levels, allowing the upper bound of expression to be determined by the promoter.

Previous work from *Chai et al., 2016* demonstrated that AcrB can be targeted for proteolysis by fusing an SsrA tag to the C-terminus. Degradation is possible because the C-terminal end of AcrB is on the cytoplasmic side of the inner membrane and can interact with cytoplasmic unfoldases (*Du et al., 2014*). Therefore, we reasoned that fusing the LOVdeg tag to AcrB would result in light-inducible degradation that disrupts activity of the AcrAB-TolC complex (*Figure 2e*). To determine whether activity of the AcrAB-TolC complex was successfully disrupted by light, we performed an antibiotic sensitivity test using chloramphenicol, which is a known substrate of the AcrAB-TolC pump (*Okusu et al., 1996*). We transformed a construct containing AcrB-LOVdeg into cells lacking the endogenous *acrB* gene (Δ*acrB*). When kept in the dark, Δ*acrB* cells with AcrB-LOVdeg showed comparable chloramphenicol tolerance to wild-type cells (*Figure 2f and g*). Blue light stimulation, in contrast, sensitized the Δ*acrB* + AcrB-LOVdeg strain to chloramphenicol compared to both wild-type and Δ*acrB* + AcrB-LOVdeg kept in the dark. This suggests that the LOVdeg tag successfully targets AcrB for degradation in a light-dependent fashion. The blue light-exposed cells still retain modest levels of chloramphenicol sensitivity when compared to Δ*acrB* cells without any AcrB complementation, likely due to basal levels of efflux pump expression from AcrB-LOVdeg relative to the knockout. As a point of comparison, we also tested chloramphenicol sensitivity of Δ*acrB* cells with IPTG-inducible AcrB expression (Δ*acrB* + AcrB) without IPTG. With no induction, this represents the lowest expression levels that can be achieved with a traditional IPTG-inducible system. The uninduced Δ*acrB* + AcrB cells displayed significantly higher tolerance to chloramphenicol when compared to wild-type and other experimental groups. This underscores the challenge of controlling potent complexes like AcrAB-TolC through the use of chemical inducers alone and demonstrates how post-translational control, such as that provided by LOVdeg, is a viable and necessary strategy to decrease expression to levels approaching those in the Δ*acrB* knockout strain.

## Tuning frequency response of the LOVdeg tag

Light-inducible systems have the potential to respond to the frequency of light exposure. Frequency-dependent tools can allow optogenetic circuits to be multiplexed beyond limited wavelength options or to add a layer of logic to optogenetic circuits (*Benzinger et al., 2022*). With added logic operations, optogenetic circuits could perform complex signal processing, analogous to those demonstrated with multiplexed chemically inducible circuits (*Shin et al., 2020*), while allowing dynamic light inputs. Additionally, higher sensitivity LOVdeg tags would also be useful in bioreactor settings, where poor light penetration into dense cultures is a feasibility concern. With these use cases in mind, we sought to characterize and alter the LOVdeg frequency response.

LOV domains utilize a flavin cofactor to absorb light. In the case of *As*LOV2, the cofactor responsible for light absorption is flavin mononucleotide (FMN). A cysteine residue in *As*LOV2 forms a reversible covalent bond with FMN, which initiates broader conformational change. A full photocycle of *As*LOV2 consists of absorption of a photon, covalent bond formation with cysteine 450, Jα helix destabilization and unfolding, decay of the cysteine-FMN bond, and Jα helix refolding (*Swartz et al., 2001*). Previous studies have determined the dynamics of bond formation and the time delay of Jα helix refolding. Further, mutations have been found that stabilize or destabilize the light state conformation (*Christie et al., 2007*; *Kawano et al., 2013*; *Zayner et al., 2013*). In the context of the LOVdeg tag, the time needed for the Jα helix to refold, known as the reversion time, likely determines

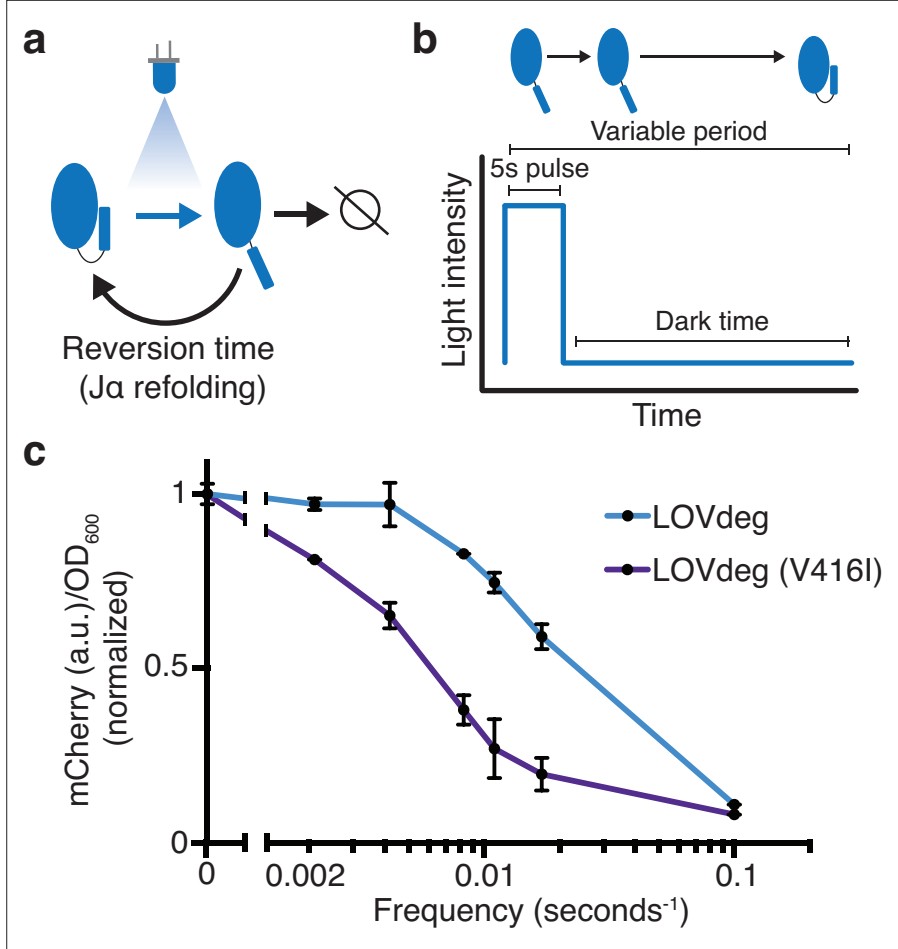

**Figure 3.** Modulating LOVdeg frequency response with photocycle mutations. (**a**) Photocycle of *AsLOV2*. Upon light absorption, the Jα helix unfolds for a period of time dictated by the stability of the light state conformation. If not degraded, the Jα helix refolds, blocking degradation. (**b**) The light program used to test frequency responses of LOVdeg photocycle variants in (**c**). A constant pulse of 5 s is followed by a variable dark time that allows for Jα helix refolding. (**c**) Expression of mCherry-LOVdeg and variant mCherry-LOVdeg (V416I) in response to different light exposure frequencies. Fluorescence values are normalized to dark state expression. Error bars show standard deviation around the mean (n = 3 biological replicates).

degradation characteristics (*Figure 3a*). In principle, the time spent in the light state dictates the amount of time the protein is susceptible to degradation and, therefore, would impact the frequency response given variable light inputs.

To test the effect of LOVdeg tag refolding dynamics, we illuminated cells with 5 s pulses of blue light followed by variable length dark periods to allow Jα helix refolding (*Figure 3b*). While holding the blue light duration fixed, we tested dark periods ranging from 475 s (5:475 s on:off, frequency of 0.002 s$^{-1}$ since there is one pulse every 480 s) to 5 s (5:5 s on:off, frequency of 0.1 s$^{-1}$) (*Figure 3b*). We used 5 s for the pulse length because it is markedly shorter than the overall degradation dynamics for the LOVdeg tag, which ensures that a single pulse is not long enough to induce significant degradation. We first tested the frequency response of the original LOVdeg tag (*Figure 3c*). The response is in line with known refolding dynamics of *AsLOV2* (*Li et al., 2020*), where over 50% degradation is only achieved at high frequencies (0.1 s$^{-1}$). Next, we tested a LOVdeg tag variant that contains a slow-photocycle mutation, V416I (*Zoltowski et al., 2009*). This amino acid substitution has been shown to increase the dark state reversion time from 8 s to 84 s in situ (*Li et al., 2020*). Indeed, for LOVdeg tag (V416I) over 50% degradation was achieved at medium frequencies (0.008 s$^{-1}$, which corresponds to 5:120 s on:off) (*Figure 3c*). This variant offers the potential for better performance in settings where increased light sensitivity is preferred, such as within bioreactors.

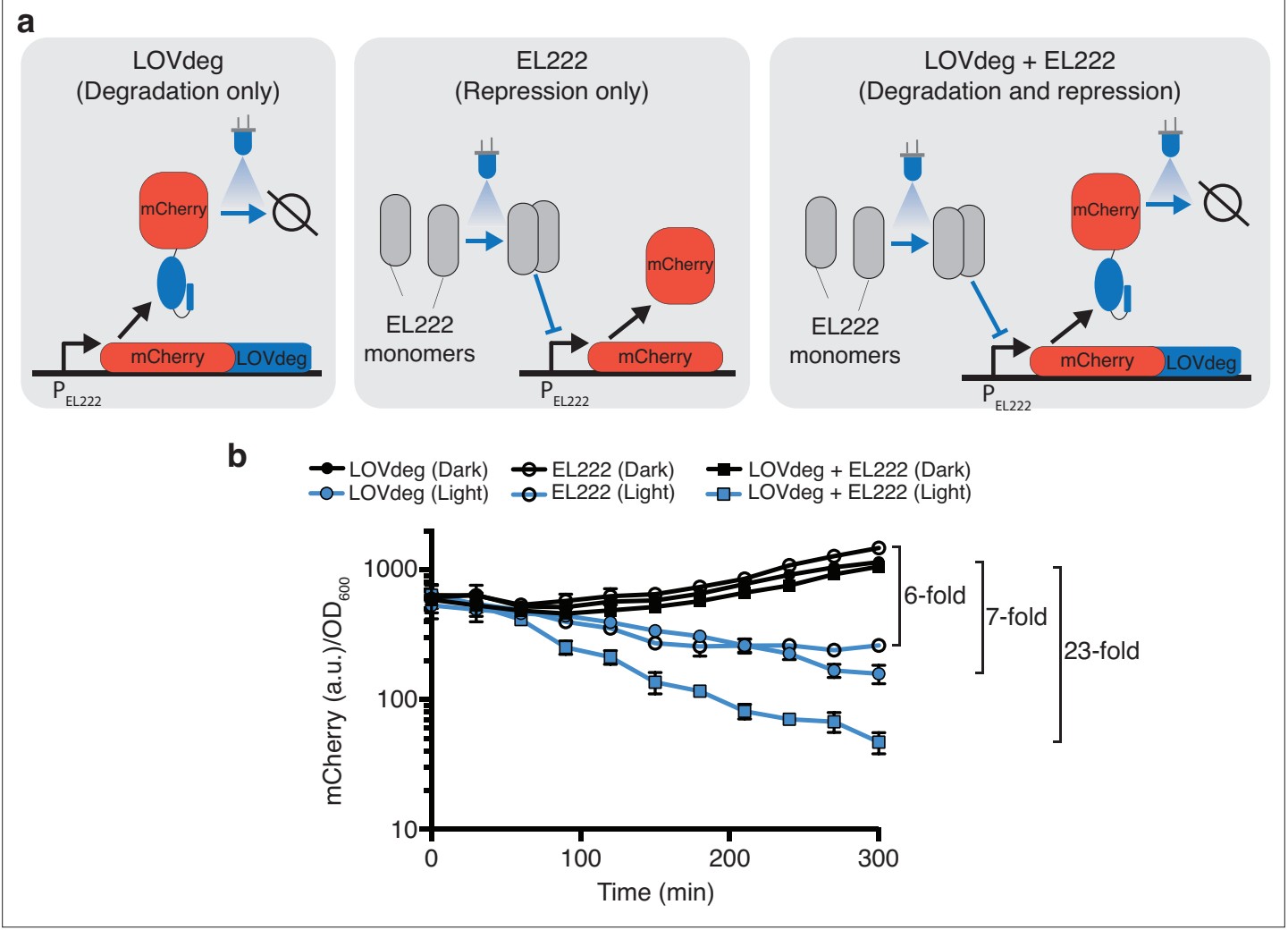

**Figure 4.** Enhanced light response using EL222 transcriptional control together with LOVdeg. (**a**) mCherry-LOVdeg expressed from an EL222-responsive promoter that is constitutively active in the absence of EL222. Addition of EL222 represses mCherry production. These two forms of regulation are combined when mCherry-LOVdeg is expressed from an EL222-responsive promoter, resulting in a circuit that both degrades and represses in response to light. (**b**) Light and dark expression of mCherry in the 'degradation only' (closed circles), 'repression only' (open circles), or 'repression + degradation' (squares) strains. Error bars show standard deviation around the mean (n = 3 biological replicates).

## Integrating LOVdeg with EL222 for multilayer control

Another attractive aspect of post-translational optogenetic control is that it can integrate with existing systems that act at the transcriptional or translational level. Adding control at multiple layers has been shown to enhance the performance and robustness of natural and synthetic systems (*Alon, 2007*; *Hasenjäger et al., 2019*; *Szydlo et al., 2022*). One commonly used system for optogenetic transcriptional control is EL222. EL222 is a blue light-responsive LOV protein that dimerizes and binds DNA upon light exposure (*Zoltowski et al., 2013*). In bacteria, EL222 can be used as a transcriptional repressor or activator depending on the placement of its binding site in the promoter (*Ding et al., 2020*; *Jayaraman et al., 2016*). We chose to combine the transcriptional repression of EL222 with the LOVdeg tag. In this arrangement, the systems work synergistically to decrease gene expression in response to blue light using simultaneous transcriptional repression and protein degradation (*Figure 4a*).

To test the performance of this combined optogenetic circuit, we created a construct in which the mCherry-LOVdeg fusion protein is driven by a promoter containing an EL222 binding site (P_{EL222}). We tested the light response of mCherry-LOVdeg expression with and without EL222 present, as well as including EL222 control of mCherry without the LOVdeg tag fused (*Figure 4b*). As expected,

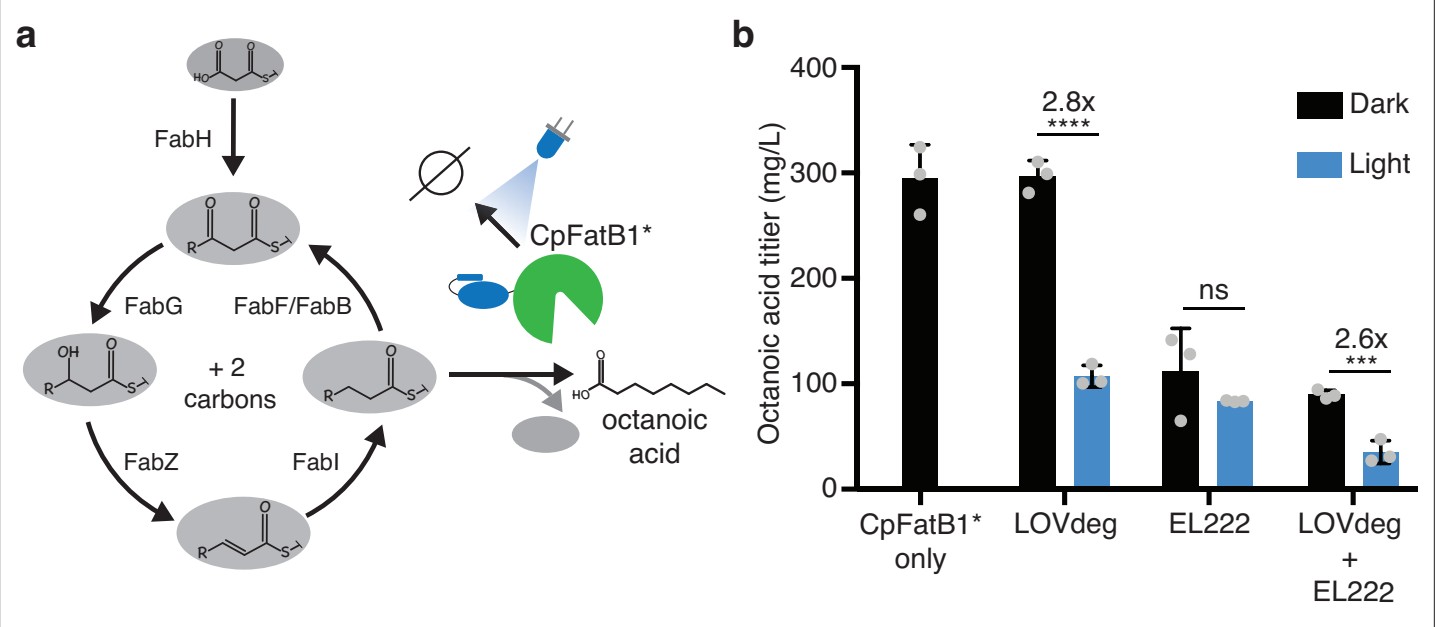

**Figure 5.** Optogenetic control of octanoic acid production. (**a**) Schematic of fatty acid synthesis in *E. coli*. *Cp*FatB1* catalyzes elongating C8-ACP molecules from this pathway to produce free octanoic acid. *Cp*FatB1* is tagged with LOVdeg to create optogenetic control. (**b**) Octanoic acid titer from strains that express *Cp*FatB1* only, *Cp*FatB1*-LOVdeg only, EL222 regulated *Cp*FatB1* only, or *Cp*FatB1*-LOVdeg+EL222. Octanoic acid is quantified by GC-MS. Strains were kept either in the dark or with continuous blue light exposure for the duration of the production period. Error bars show standard deviation around the mean (****p<0.0001; ***p<0.0001; ns, not significant; two-tailed unpaired *t*-test; n = 3 biological replicates).

mCherry-LOVdeg expression decreased in response to light even without EL222 present, representing the sole action of the degradation tag. Similarly, EL222-only mCherry decreased in response to light due to repression by EL222. However, multilayer control resulted in a faster decrease in expression in response to light and reached lower levels compared to either LOVdeg or EL222 alone. The fold change decrease in expression was improved from 7× and 6× for just the LOVdeg or EL222, respectively, to 23× when both systems were combined.

## Optogenetic control of octanoic acid production

We next sought to apply the LOVdeg tag to a metabolic engineering task as we envision that post-translational control will be especially advantageous in these biotechnology applications. Transcriptional control alone is particularly problematic in a metabolic engineering setting because chemical production is typically carried out at stationary phase with slow growth rates, meaning that proteins expressed at basal levels will accumulate in production settings. Therefore, dynamic control using transcriptional optogenetic systems alone will only allow protein levels to increase or plateau. However, a prerequisite for dynamic control of metabolic pathways is that enzyme levels can be modulated to turn off production.

As a proof of concept, we chose to control the enzyme *Cp*FatB1 with LOVdeg, EL222, or the EL222-LOVdeg circuit. *Cp*FatB1 is an acyl-ACP thioesterase from *Cuphea palustris* that primarily catalyzes octanoyl-ACP to produce octanoic acid. Octanoic acid is a valuable medium-chain oleo-chemical with limited natural sources (*Sarria et al., 2017*). Specifically, we expressed the catalytically enhanced mutant from Hernandez-Lozada et al., *Cp*FatB1.2-M4-287, which we denote here as *Cp*FatB1* (*Hernández Lozada et al., 2018*). This enzyme interfaces with the endogenous fatty acid synthesis pathway in *E. coli* to produce free fatty acids (*Figure 5a*). In this pathway, the carbon tail of an acyl-ACP moiety is elongated two carbons at a time. *Cp*FatB1* specifically catalyzes C8-ACPs, which results in production of free octanoic acid. Importantly, since *Cp*FatB1* is very catalytically active, low levels of expression are optimal for production and strains with high expression exhibit a growth defect (*Hernández Lozada et al., 2018*). Therefore, to dynamically regulate *Cp*FatB1* activity in cells, expression must be controlled in a low range, which is particularly challenging in stationary phase.

A first step toward optogenetically controlling this metabolic pathway is to demonstrate that enzyme levels can be modulated between protein levels that are relevant to endpoint titers. If expression can only oscillate between high and medium protein levels, dynamic light control will not be effective. To test whether we could effectively stunt $Cp$FatB1* activity, we controlled enzyme expression using only LOVdeg tag degradation, only EL222 repression, or combined LOVdeg and EL222 throughout at 24 hr fermentation period (*Figure 5b*). LOVdeg alone successfully decreased octanoic acid in the light condition. The EL222-only condition did not significantly decrease octanoic acid production with light compared to conditions in the dark, demonstrating the shortcomings of solely transcriptional control at stationary phase. In contrast, the LOVdeg tag with EL222 resulted in a significant decrease in octanoic acid in the continuous light condition. Thus, protein degradation is needed to effectively shunt the metabolic pathway during stationary phase.

Interestingly, the LOVdeg-only strain, in which $Cp$FatB1*-LOVdeg is expressed through the IPTG-inducible LacUV5 promoter, exhibits higher dark state octanoic acid production compared to $Cp$FatB1* expressed under the EL222 promoter. The dark state production of the LOVdeg-only strain is similar to that of $Cp$FatB1* without the LOVdeg tag, both expressed via the same LacUV5 promoter. This is likely due to limited constitutive expression from $P_{EL222}$. This limitation further demonstrates the utility of post-translational control where the promoter is not involved in the light response. Previous efforts aimed at increasing the strength of EL222 responsive promoters have been carried out (*Ding et al., 2020*), and further promoter engineering would be needed to increase EL222 promoter strength to increase octanoic acid production in the dark state. Clearly, decreasing titers is not the overall goal in metabolic engineering. However, the ability to control expression at relevant ranges throughout stationary phase represents an important stepping stone to investigate dynamic control schemes or feedback control that may lead to enhanced strain performance.

## Discussion

Here, we developed and characterized the LOVdeg tag, which provides blue light-inducible protein degradation, offering unique advantages as a tool for bacterial synthetic biology. Protein degradation offers a mode of control that is currently limited in the bacterial optogenetic toolkit. The bacterial light-responsive degradation system from *Komera et al., 2022* accomplishes protein degradation indirectly through an exogenous split protease. In comparison, in our approach the degradation tag itself is light responsive and does not require extra components, making it straightforward to incorporate. In this study, we showed that the truncated $As$LOV2 protein can be fused to the C-terminal end of various proteins to destabilize them and then target them for degradation. The addition of stabilizing mutations from iLID (to create $As$LOV2*) results in low basal activity in the dark, with strong switching in response to light. Aside from creating a light-responsive degradation tag, the effectiveness of these stabilizing mutations may serve to improve the switch-like behavior in many other $As$LOV2-based systems.

We demonstrated the modularity of the LOVdeg tag by incorporating it into three distinct contexts with little to no fine-tuning. Each system, the LacI repressor, CRISPR activator, and the AcrB efflux pump, was converted to optogenetically controlled by simply tagging the protein with LOVdeg. This modularity mimics the ease of use of transcriptional optogenetic systems, which only require a change of the promoter. However, incorporating light control at the protein level simplifies this process because protein levels can remain in their native, or previously fine-tuned, context. This configuration is especially beneficial for proteins that require low expression as transcriptional optogenetic systems often suffer from leaky basal expression. The LOVdeg tag provides a design that can work with endogenous protein levels, circumventing this issue. In this study, we focused on proteins encoded on plasmids expressed from synthetic promoters. However, we anticipate the LOVdeg tag can be useful in systems biology contexts where natural promoters linked to endogenous gene networks are best left untouched. In this case, the LOVdeg tag can add the ability to optogenetically manipulate genes while keeping them in their native gene regulatory context.

Since degradation occurs post-translationally, the LOVdeg tag can easily be integrated with existing optogenetic systems to enhance their function. We demonstrated this by combining the LOVdeg tag with the EL222 repression system, showing that the synergistic action of transcriptional repression and degradation results in an increase in the dynamic range, owing to the lower off-state under light illumination. In this configuration, EL222 and the LOVdeg tag work coherently to decease expression.

However, the LOVdeg tag can potentially be incorporated into activating systems incoherently to create dynamic functionality, such as pulse generators and inverters (*Benzinger et al., 2022*).

Throughout this study, we performed experiments using low copy number plasmids that result in moderate levels of a given protein of interest. While this is a common use case for synthetic biology, degradation as the sole mode of gene expression control may be limiting when proteins are at very high levels. As an ATP-dependent process that utilizes a finite pool of proteolytic machinery, degradation rates can saturate at sufficiently high protein levels (*Cookson et al., 2011*). Furthermore, using protein degradation as the sole mode of gene expression control is wasteful in some contexts. Constitutive transcription and translation of a gene followed by degradation utilizes valuable cellular resources, which is an important consideration in metabolic engineering. To avoid this type of energetic waste, we envision the LOVdeg tag could be used in concert with other modes of regulation, as we demonstrated with EL222. In addition, it is unclear whether the LOVdeg tag is compatible with other prokaryotic (including mitochondrial) proteasomes. Future studies focused on the portability of the LOVdeg tag may address this question and potentially lead to further mechanistic insight.

In summary, the LOVdeg tag offers a straightforward route for introducing optogenetic control of protein degradation in *E. coli*. By lowering the barrier to entry for incorporating light responsiveness into a protein of interest, we envision that systems typically studied with chemical induction or constitutive expression can now be controlled optogenetically without extensive fine-tuning. Furthermore, the LOVdeg tag can act as a circuit enhancer when incorporated into existing optogenetic systems to increase functionality and robustness.

# Materials and methods
## Strains and plasmids
We used *E. coli* BW25113 as the wild-type strain. All knockout strains are from the Keio collection (*Baba et al., 2006*), which were derived from BW25113. We used Golden Gate cloning to create all plasmid constructs (*Engler et al., 2008*; *Supplementary file 1*). The IPTG-inducible constructs were derived from pBbS5c-mRFP1 from the BglBrick plasmid library (*Lee et al., 2011*). In the constitutive version of the mCherry-AsLOV2 variants, we swapped the IPTG-inducible promoter with a constitutive promoter, $P_{W7}$ (5'-ttatcaaaaagagtattgaaataaagtctaacctataggaagattacagccatcgagagggacacggcgaa-3'). We used this constitutive version for microscopy and the protease knockout studies; all other experiments used the IPTG-inducible promoter $P_{lacUV5}$.

In all cases of protein fusions to *As*LOV2 variants, a five amino acid GS linker of 'S-S-G-S-G' was used between the protein of interest and the LOVdeg tag. Sequences for the *As*LOV2 variants are provided in *Supplementary file 3*. In the LacI-LOVdeg experiments, pBbS5c-mCherry was also used as the backbone, but with the LOVdeg sequence cloned after the *lacI* gene instead of *mCherry*. In the LacI-LOVdeg experiment to test the impact of basal LacI, pBbS5c-LacI-LOVdeg-mCherry was co-transformed with pBbAa-LacI-Decoy from *Wang et al., 2021*. CRISPRa constructs were based on the MS2-SoxS system from *Dong et al., 2018*. The J106 gRNA sequence from Dong et al. was used to target dCas9 upstream mRFP1 under control of the minimal J1 promoter. This gRNA plasmid was constructed using the Golden Gate assembly method to replace the targeting sequence in the pCD061 backbone (Addgene #113315). The mRFP1 reporter plasmid was derived from pJF076Sa (Addgene #113322) by replacing the ampicillin resistance gene with a kanamycin resistance gene from the BglBrick library. MS2-SoxS-LOVdeg was expressed from a variant of pJF093 (Addgene #113323). TetR and its corresponding promoter driving expression of MS2-SoxS were replaced with the LacI-$P_{Trc}$-inducible system from pBbA1c-mRFP1 from the BglBrick library (*Lee et al., 2011*). The inducer was changed to IPTG because aTc is sensitive to blue light (*Baumschlager et al., 2020*). The LOVdeg tag was added to this construct with a C-terminal fusion to SoxS. Plasmids containing *acrAB* were built from pBbA5k-acrAB from *El Meouche and Dunlop, 2018*. The FLP recombination protocol from Datsenko and Wanner was used to cure the *kanR* cassette from the genome of the Δ*acrB* strain (Keio collection, JW0451) (*Datsenko and Wanner, 2000*).

The slow photocycle variant of the LOVdeg tag, V416I, was constructed using site-directed mutagenesis of Valine at amino acid position 416.

EL222 was synthesized by IDT (Coralville, IA) and plasmids were constructed to mimic EL222 repression systems from *Jayaraman et al., 2016* (*Supplementary file 1*). A variant of the promoter

$P_{BLrep-v1}$ from Ding et al., $P_{raB}$ was used in all EL222 experiments, and we refer to it in figures as $P_{EL222}$ (**Ding et al., 2020**). Plasmid pBbE5k-$P_{EL222}$-mCherry-LOVdeg was co-transformed with pEL222, which constitutively expresses EL222 (**Supplementary file 2**).

For octanoic acid production experiments, the coding sequence of *Cp*FatB1.2-M4-287 derived from Hernández Lozada et al. was synthesized by Twist Biosciences (South San Francisco, CA) and cloned after the $P_{EL222}$ or $P_{lacUV5}$ promoter (**Hernández Lozada et al., 2018**). Plasmid pBbE5k-$P_{EL222}$-*Cp*FatB1*-LOVdeg was co-transformed with pEL222 for LOVdeg + EL222 production control (**Supplementary file 2**).

Plasmids expressing ClpA, ClpX, and HslU were constructed by amplifying the unfoldase gene from the wild-type genome and inserting it into the pBbA8k backbone from the BglBrick library (**Lee et al., 2011**).

Constructs from this work are available on AddGene: https://www.addgene.org/Mary_Dunlop/.

## Blue light stimulation

Unless otherwise noted, bacteria were cultured in Luria broth (LB) with appropriate antibiotics for plasmid maintenance at 37°C with 200 rpm shaking. Antibiotic concentrations used for plasmid maintenance were 30 µg/mL for kanamycin, 100 µg/mL for carbenicillin, and 25 µg/mL for chloramphenicol. All light exposure experiments were carried out with a light plate apparatus (LPA) (**Gerhardt et al., 2016**) using 465 nm blue light. Overnight cultures of light-sensitive strains were diluted 1:50 and precultured in the dark for 2 hr. For IPTG-inducible constructs, 1 mM IPTG was added when the cells were diluted. After 2 hr in the dark, cells were exposed to blue light in the LPA at a setpoint of 100 µW/cm². Red fluorescence (excitation 560 nm, emission 600 nm) and optical density (OD) readings were taken using a BioTek Synergy H1m plate reader (BioTek, Winooski, VT) after 5 hr of incubation unless otherwise noted. For experiments testing degradation at various growth phases, a 4 hr light window was introduced at variable times, as shown in **Figure 1**. For CRISPRa experiments, light stimulation was continued for 8 hr prior to RFP and OD readings. For frequency response experiments, the LPA was programmed using its Iris software (https://taborlab.github.io/Iris/; **Gerhardt, 2016**) to pulse blue light at varying frequencies. A 5 s light pulse was kept constant for each experiment while the time between pulses was varied (5 s, 55 s, 85 s, 115 s, 235 s, and 480 s).

## Chloramphenicol sensitivity testing

Chloramphenicol sensitivity experiments were performed in M9 minimal media (M9 salts, 2 mM MgSO₄, 100 µM CaCl₂) with 1% glucose at 37°C with 200 rpm shaking. Overnight cultures in LB were initially diluted 1:50 into M9 media for 4 hr. The M9 conditioned cultures were then diluted again 1:20 into 24-well plates containing M9 media with varying levels of chloramphenicol (0, 0.3125, 0.625, 1.25, 2.5, 5, and 10 µg/mL) and grown for 20 hr before measuring OD using a BioTek Synergy H1m plate reader (BioTek). We also conducted experiments with the *E. coli* BW25113 Δ*acrB* strain (referred to as Δ*acrB*). For these, Δ*acrB* was transformed with pBbA5k-AcrAB-LOVdeg (**Supplementary file 2**) and chloramphenicol sensitivity experiments were carried out either in the LPA with constant light illumination or kept in the dark for the duration of growth. No IPTG was added to the ΔacrB + AcrAB-LOVdeg cultures because basal expression was enough to recover wild-type resistance. The same chloramphenicol sensitivity protocol was performed in the dark using wild-type BW25113, Δ*acrB* + AcrB without induction, and Δ*acrB* as controls.

## Microscopy

Strains were grown overnight in LB medium. Cultures were refreshed 1:100 in M9 minimal media for microscopy (M9 salts supplemented with 2 mM MgSO₄, 0.1 mM CaCl₂) with 0.4% glucose for 2 hr. Samples were then placed on 1.5% low melting agarose pads made with M9 minimal media for microscopy with 0.4% glucose. Samples were grown at 30°C. Cells were imaged at 100× using a Nikon Ti-E microscope. Blue light exposure was provided by a LED ring (Adafruit NeoPixel 1586), which was fixed above the microscope stage and controlled by an Arduino using a custom MATLAB script. Blue light was kept constant except during image acquisitions.

## Octanoic acid production experiment

For octanoic acid production experiments, strains expressing *Cp*FatB1* under various modes of control (**Supplementary file 2**) were cultured in LB overnight with light illumination to maintain low *Cp*FatB1* expression. Overnight cultures were diluted 1:20 into M9 minimal media with 2% glucose and kept in the light until they reached early stationary phase ($OD_{600}$ of 0.6) unless otherwise noted. The LPA was then programmed to either maintain light for low octanoic acid production or turn off light exposure to induce octanoic acid production for 24 hr prior to fatty acid extraction and quantification.

## Fatty acid quantification

Samples for GC-MS quantification were taken at 24 hr. Also, 400 µL of vortexed culture was used for fatty acid extraction and derivatization into fatty acid methyl esters as described by *Sarria et al., 2018* with the following minor modifications: an internal standard of nonanoic acid (C9) was added to the 400 µL sample at a final concentration of 88.8 mg/L and vortexed for 5 s. The following was then added to the sample for fatty acid extraction and vortexed for 30 s: 50 µL 10% NaCl, 50 µL glacial acetic acid, and 200 µL ethyl acetate. The sample was then centrifuged at 12,000 × *g* for 10 min. After centrifugation, 100 µL of the ethyl acetate layer was mixed with 900 µL of a 30:1 mixture of methanol:HCl (12 N) in a 2 mL microcentrifuge tube. The solution was vortexed for 30 s followed by an incubation at 50°C for 60 min for methyl ester derivatization. Once cooled to room temperature, 500 µL hexanes and 500 µL water were added to the 2 mL microcentrifuge tube, vortexed for 10 s, and allowed to settle. Then, 250 µL of the hexane layer was mixed with 250 µL ethyl acetate in a GC-MS vial for quantification.

The samples were analyzed with an Agilent 6890N/Agilent 5973 MS detector using a DB-5MS column. The inlet temperature was set to 300°C with flow at 4 mL/min. The oven heating program was initially set to 70°C for 1 min, followed by a ramp to 290°C at 30°C/min, and a final hold at 290°C for 1 min. GLC-20 and GLC-30 FAME standard mixes (Sigma) were tested using this protocol to ensure proper capture of all chain lengths and to gauge retention times. Internal standards were used for quantification, with chain lengths C8-C12 quantified with the nonanoic acid internal standard and C14-C18 quantified with the pentadecanoic internal standard.

## Acknowledgements

We thank members of the Dunlop Lab for helpful discussions. This work was supported by DOE grant DE-SC0019387, NSF grant CBET-1804096, and NIH grant R01AI102922. CCO and MBS received support through the NIH training grants T32 GM130546 and T32 EB006359, respectively.

## Additional information

### Funding

| Funder | Grant reference number | Author |
| --- | --- | --- |
| Department of Energy | DE-SC0019387 | Mary J Dunlop |
| National Science Foundation | CBET-1804096 | Mary J Dunlop |
| National Institutes of Health | R01AI102922 | Mary J Dunlop |
| National Institutes of Health | T32 GM130546 | Cristian Coriano-Ortiz |
| National Institutes of Health | T32 EB006359 | Michael B Sheets |

The funders had no role in study design, data collection and interpretation, or the decision to submit the work for publication.

## Author contributions

Nathan Tague, Conceptualization, Data curation, Software, Formal analysis, Validation, Investigation, Visualization, Methodology, Writing – original draft, Writing – review and editing; Cristian Coriano-Ortiz, Validation, Investigation, Writing – review and editing; Michael B Sheets, Software, Validation; Mary J Dunlop, Conceptualization, Supervision, Funding acquisition, Writing – original draft, Project administration, Writing – review and editing

## Author ORCIDs

Nathan Tague ⬡ http://orcid.org/0000-0002-8114-6700
Mary J Dunlop ⬡ https://orcid.org/0000-0002-9261-8216

Reviewer #1 (Public Review): https://doi.org/10.7554/eLife.87303.3.sa1
Reviewer #2 (Public Review): https://doi.org/10.7554/eLife.87303.3.sa2
Reviewer #3 (Public Review): https://doi.org/10.7554/eLife.87303.3.sa3
Author Response https://doi.org/10.7554/eLife.87303.3.sa4

# Additional files

## Supplementary files

- Supplementary file 1. Plasmids used in this study.
- Supplementary file 2. Strains used in this study.
- Supplementary file 3. DNA sequences used in this study.
- MDAR checklist

## Data availability

All source data associated with the manuscript are available via Zenodo at https://doi.org/10.5281/zenodo.10439843.

The following dataset was generated:

| Author(s) | Year | Dataset title | Dataset URL | Database and Identifier |
|---|---|---|---|---|
| Tague N, Coriano-Ortiz C, Sheets MB, Dunlop MJ | 2024 | Light inducible protein degradation in *E. coli* with the LOVdeg tag | https://zenodo.org/records/10439844 | Zenodo, 10.5281/zenodo.10439843 |

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

## Appendix 1

### Mechanistic insights into the *E. coli* proteasome

Based on the exposed C-terminal amino acids in the LOVdeg (E-A-A), we anticipated that degradation would be primarily mediated by ClpXP because the C-terminal alanines, when on an unstructured peptide of sufficient length, are known to be adequate for ClpX targeting (*Fei et al., 2020*). However, degenerate degradation from multiple endogenous proteases is common in *E. coli* and has been demonstrated with SsrA-tagged substrates (*Flynn et al., 2001*). Additionally, ClpX is the most extensively studied unfoldase, meaning the rules governing other endogenous unfoldases are less understood and their action should not be ruled out. ClpA, ClpX, and HslU unfoldases utilize protease counterparts to perform protein degradation with ClpAP, ClpXP, and HslVU complexes, respectively. In contrast, Lon performs both unfoldase and protease activity. To understand the endogenous unfoldase(s) responsible for degradation of the LOVdeg tag, we expressed the mCherry-LOVdeg construct in several unfoldase knockouts, including those deleting the genes that encode ClpA, ClpX, HslU, or Lon (*Figure 1—figure supplement 3a*). FtsH, the last of the five unfoldase-proteases in *E. coli*, was not included in the knockout study because it is an essential gene and could not be knocked out.

We measured reductions in mCherry in response to blue light induction in the different knockout backgrounds. In each knockout, we expressed mCherry-LOVdeg under a constitutive promoter. We also expressed the non-truncated counterpart for a degradation-resistant comparison. Protein expression was decreased in light for all knockout strains; however, the degree of reduction varied (*Figure 1—figure supplement 3b*). The fold change of degradation was reduced relative to wild type in Δ*clpA*, Δ*clpX*, and Δ*hslU* strains (*Figure 1—figure supplement 3c*). To further test which of these unfoldases was responsible for LOVdeg tag degradation, we created plasmids expressing each unfoldase exogenously under an arabinose-inducible promoter to see whether excess unfoldase would increase degradation of mCherry. ClpA was the only one to display increased degradation when overexpressed (*Figure 1—figure supplement 3d*). With ClpA expressed from plasmid in addition to endogenous ClpA, the half-life of mCherry-LOVdeg was decreased from 74 to 38 min. Given the fold change decrease we observed in Δ*hslU*, we were surprised that strains overexpressing HslU did not increase their degradation rate (*Figure 1—figure supplement 4*). The lack of enhanced degradation when HslUV is overexpressed suggests that it is not the primary source of LOVdeg tag degradation. It is possible that the decreased degradation fold change seen in Δ*hslU* can be attributed to broader systemic changes in this knockout strain. Although the Δ*clpX* strain showed a reduced fold change, this is likely due to generalized changes in expression, which we observed with mCherry with no degradation tag in this strain as well (*Figure 1—figure supplement 5*). Our initial assumption that ClpXP would be the primary source of degradation was incorrect. The data instead show that ClpA is involved in LOVdeg tag degradation; however, it is likely a single unfoldase-protease is not entirely responsible for degradation.

Since ClpA was implicated in degradation, we also tested knockouts of ClpP and ClpS. ClpP is responsible for proteolysis of substrates unfolded by ClpA, and ClpS is an adaptor protein for ClpAP that alters targeting specificity (*Baker and Sauer, 2006*). mCherry was degraded efficiently in both the Δ*clpP* and Δ*clpS* strains (*Figure 1—figure supplement 6*). Because ClpA is implicated in degradation, we initially expected ClpP, the protease counterpart to ClpA, would be necessary. However, studies examining degradation of SsrA-tagged substrates have shown that substrates can still be degraded efficiently, even in Δ*clpP* strains (*Farrell et al., 2005*; *Lies and Maurizi, 2008*). For example, Lies et al. found that SsrA-tagged substrates can be degraded in Δ*clpP* strains but accumulate in Δ*clpP* Δ*lon* strains. They concluded that in the absence of ClpP, ClpA and ClpX continue to unfold substrates and Lon carries out proteolysis on the unfolded substrates. A similar mechanism may be at play with the LOVdeg tag in Δ*clpP* cells.

By knocking out exogenous *E. coli* unfoldases, we gained partial insight into the mechanism of LOVdeg tag destabilization. The Δ*clpA* knockout exhibits decreased degradation in response to light, while complementing cells with *clpA* increases degradation speed, demonstrating the involvement of the ClpA protease in LOVdeg tag destabilization. However, other proteolytic activity is also involved, as degradation could still be achieved, albeit to a lesser extent, in the Δ*clpA* strain. Further, full degradation was maintained in its partner Δ*clpP* strain. It remains unclear whether the LOVdeg tag is targeted due to specific amino acid sequence interactions with a given unfoldase or if

the general disorder induced at the C-terminal end of the protein is sufficient for recognition by the proteasome. The non-truncated versions of *AsLOV2* and *AsLOV2\**, which maintain light-dependent C-terminal disorder, are stable, suggesting that it is a mix of sequence and C-terminal peptide disorder. The full mechanism of LOVdeg tag destabilization is a topic for future investigation.

## Adding decoy sites to reduce impact of basal LacI does not improve LacI-LOVdeg response

We sought to further investigate why LacI-LOVdeg responded with higher mCherry expression under IPTG induction than light induction. One possibility is that low levels of LacI are escaping degradation and causing basal repression. To address this, we added LacI decoy binding sites that work by binding excess LacI, following a method we developed in a previous study (*Wang et al., 2021*). With the decoy present, light-induced expression was only slightly increased (*Figure 2— figure supplement 1*). This indicates that low levels of LacI are not the primary explanation for the discrepancy between IPTG induction and light induction. We hypothesize that the lowered expression with light exposure stems from the time delay inherent in protein degradation compared to allosteric binding of IPTG to LacI.

