## [Editor Report · eLife assessment]

This **valuable** study reports on a new tool that allows for light-controlled protein degradation in *Escherichia coli*. With the improved light-responsive protein tag, endogenous protein levels can be reduced several fold. The methodology is **convincing** and will be of interest to the fields of gene expression regulation in bacteria and more generally to synthetic biologists.

---

## [Referee Report · Reviewer #1 (Public Review)]

Specifically controlling the level of proteins in bacteria is an important tool for many aspects of microbiology, from basic research to protein production. While there are several established methods for regulating transcription or translation of proteins with light, optogenetic protein degradation has so far not been established in bacteria. In this paper, the authors present a degradation sequence, which they name "LOVdeg", based on iLID, a modified version of the blue-light-responsive LOV2 domain of Avena sativa phototropin I (AsLOV2). The authors reasoned that by removing the three C-terminal amino acids of iLID, the modified protein ends in "-E-A-A", similar to the "-L-A-A" C-terminus of the widely used SsrA degradation tag. The authors further speculated that, given the light-induced unfolding of the C-terminal domain of iLID and similar proteins, the "-E-A-A" C-terminus would become more accessible and, in turn, the protein would be more efficiently degraded in blue light than in the dark.

Indeed, several tested LOVdeg-tagged proteins show clearly lower cellular levels in blue light than in the dark. Depending on the nature and expression level of the target protein, protein levels are reduced modestly to strongly (2 to 20x lower levels upon illumination). Accordingly, the authors propose to use their system in combination with other light-controlled expression systems and provide data validating this approach. The LOVdeg system allows to modulate protein levels to a similar degree and with comparable kinetics as optogenetic systems controlling transcription or translation of protein, and can be combined with such systems.

The manuscript and the figures are generally very well-composed and follow a clear structure. The schematics nicely explain the underlying principles. Besides the advantages of the LOVdeg approach, including its complementarity to controlled expression of proteins, the revised version of the manuscript also highlights the limitations of the method more clearly, e.g., (i) the need to attach a C-terminal tag of considerable size to the protein of interest, (ii) the limited efficiency (slightly less efficient and slower than EL222, a light-dependent transcriptional control mechanism), and (iii) the incompletely understood prerequisites for its application. Taken together, this manuscripts describes the LOVdeg system as a valuable addition to the tool box for controlling protein levels in prokaryotic cells.

---

## [Referee Report · Reviewer #2 (Public Review)]

In this manuscript the authors present and characterize LOVdeg, a modified version of the blue-light sensitive AsLOV2 protein, which functions as a light-inducible degron in *Escherichia coli*. Light has been shown to be a powerful inducer in biological systems as it is often orthogonal and can be controlled in both space and time. Many optogenetic systems target regulation of transcription, however in this manuscript the authors target protein degradation to control protein levels in bacteria. This is an important advance in bacteria, as inducible protein degradation systems in bacteria have lagged behind eukaryotic systems due to protein targeting in bacteria being primarily dependent on primary amino acid sequence and thus more difficult to engineer. In this manuscript, the authors exploit the fact that the J-alpha helix of AsLOV2, which unwinds into a disordered domain in response to blue light, contains an E-A-A amino acid sequence which is very similar to the C-terminal L-A-A sequence in the SsrA tag which is targeted by the unfoldases ClpA and ClpX. They truncate AsLOV2 to create AsLOV2(543) and combine this truncation with a mutation that stabilizes the dark state to generate AsLOV2*(543) which, when fused to the C-terminus of mCherry, confers light-induced degradation. The authors do not verify the mechanism of degradation due to LOVdeg, but evidence from deletion mutants contained in the supplemental material hints that there is a ClpA dominated mechanism. The LOVdeg is able to target mCherry for protein degradation across different phases of bacterial growth, which is important for regulating processes at stationary phase and a potential additional advantage over transcriptional repression systems. They demonstrate modularity of this LOVdeg by using it to degrade the LacI repressor, CRISPRa activation through degradation of MCP-SoxS, and the AcrB protein which is part of the AcrAB-TolC multidrug efflux pump. In all cases, measurement of the effect of the LOVdeg is indirect as the authors measure reduction in LacI repression, reduction in CRISPRa activation, and drug resistance rather than directly measuring protein levels. Nevertheless the evidence is convincing, although seemingly less effective than in the case of mCherry degradation, although it is hard to compare due to the different endpoints being measured. The authors further modify LOVdeg to contain a known photocycle mutation that slows its reversion time in the dark, so that LOVdeg is more sensitive to short pulses of light which could be useful in low light conditions or for very light sensitive organisms. They also demonstrate that combining LOVdeg with a blue-light transcriptional repression system (EL222) can decrease protein levels an additional 23-fold (relative to 7-fold with LOVdeg alone). Finally, the authors apply LOVdeg to a metabolic engineering task, namely reducing expression of octanoic acid by regulating the enzyme CpFatB1, an acyl-ACP thioesterase. The authors show that tagging CpFatB1 with LOVdeg allows light induced reduction in octanoic acid titer over a 24 hour fermentation. In particular, by comparing control of CpFatB1 with EL222 transcriptional repression alone, LOVdeg, or both the authors show that light-induced protein degradation is more effective than light-induced transcriptional repression. The authors suggest that this is because transcriptional repression is not effective when cells are at stationary phase (and thus there is no protein dilution due to cell division). Overall, the authors have generated a modular, light-activated degron tag for use in Escherichia coli that is likely to be a useful tool in the synthetic biology and metabolic engineering toolkit.

---

## [Referee Report · Reviewer #3 (Public Review)]

The authors present the mechanism, validation, and modular application of LOVtag, a light-responsive protein degradation tag that is processed by the native degradosome of *Escherichia coli*. Upon exposure to blue light, the c-terminal alpha helix unfolds, essentially marking the protein for degradation. The authors demonstrate the engineered tag is modular across multiple complex regulatory systems, which shows its potential widespread use throughout the synthetic biology field. The step-by-step rational design of identifying the protein that was most dark-stabilized as well as most light-responsive for degradation, was useful in terms of understanding the key components of this system. The most compelling data shows that the engineered LOVTag can be fused to multiple proteins and achieve light-based degradation, without affecting the original function of the fused protein.

---

## [Author Response]

The following is the authors’ response to the original reviews.

Note to all Reviewers

We appreciate the reviewers’ comments and suggestions for improving the manuscript. Below is a summary of new data added and a brief description of the major new results. A detailed pointby-point response follows.

New data:

• Figure 1f

• Figure 2b, f, g

• Figure 4b

• Figure S7 • Figure S8

• Figure S9

Summary of major new results/edits:

• At the request of Reviewer #1 we have updated the name of the degradation tag to be more specific and we now call it the “LOVdeg” tag.

• We have added new controls demonstrating that light stimulation does not cause photobleaching or toxicity issues (Fig. S7).

• We now show that LOVdeg can function at various points in the growth cycle, demonstrating robust degradation (Fig. 1f, Fig. S8).

• We have included relevant controls for the AcrB-LOVdeg efflux pump results (Fig. 2f-g).

• We have included important benchmarking controls, such as an EL222-only control and SsrA tag control to provide a clearer view of how LOVdeg performance compares to other systems (Fig. S9, Fig. 4b).

Additional note:

• While repeating experiments during the revision process we found that the results for the combined action of EL222 and the LOVdeg tag were not as dramatic as in our original measurements, though the overall findings are consistent with our original results. Specifically, we still find that the combination of EL222 and the LOVdeg tag produces a lower signal than either on their own. We have updated these data in the revised manuscript (Fig. 4b).

**Reviewer #1:**

**Public Review:**
Specifically controlling the level of proteins in bacteria is an important tool for many aspects of microbiology, from basic research to protein production. While there are several established methods for regulating transcription or translation of proteins with light, optogenetic protein degradation has so far not been established in bacteria. In this paper, the authors present adegradation sequence, which they name "LOVtag", based on iLID, a modified version of the blue-light-responsive LOV2 domain of Avena sativa phototropin I (AsLOV2). The authors reasoned that by removing the three C-terminal amino acids of iLID, the modified protein ends in "-E-A-A", similar to the "-L-A-A" C-terminus of the widely used SsrA degradation tag. The authors further speculated that, given the light-induced unfolding of the C-terminal domain of iLID and similar proteins, the "-E-A-A" C-terminus would become more accessible and, in turn, the protein would be more efficiently degraded in blue light than in the dark.Indeed, several tested proteins tagged with the "LOVtag" show clearly lower cellular levels in blue light than in the dark. While the system works efficiently with mCherry (10-20x lower levels upon illumination), the effect is rather modest (2-3x lower levels) in most other cases. Accordingly, the authors propose to use their system in combination with other light-controlled expression systems and provide data validating this approach. Unfortunately, despite the claim that the "LOVtag" should work faster than optogenetic systems controlling transcription or translation of protein, the degradation kinetics are not consistently shown; in the one case where this is done, the response time and overall efficiency are similar or slightly worse than for EL222, an optogenetic expression system.The manuscript and the figures are generally very well-composed and follow a clear structure. The schematics nicely explain the underlying principles. However, limitations of the method in its main proposed area of use, protein production, should be highlighted more clearly, e.g., (i) the need to attach a C-terminal tag of considerable size to the protein of interest, (ii) the limited efficiency (slightly less efficient and slower than EL222, a light-dependent transcriptional control mechanism), and (iii) the incompletely understood prerequisites for its application. In addition, several important controls and measurements of the characteristics of the systems, such as the degradation kinetics, would need to be shown to allow a comparison of the system with established approaches. The current version also contains several minor mistakes in the figures.

We thank reviewer #1 for the feedback and suggestions to strengthen the manuscript. We have addressed these comments in the points that follow and now include important controls and benchmarks for our molecular tool.

Major points1. The quite generic name "LOVtag" may be misleading, as there are many LOV-based tags for different purposes.

We appreciate that it would be beneficial to have a more specific name. We have updated the name to “LOVdeg” tag, which captures both the inclusion of LOV and the degradation function of the tag.

Updated throughout the manuscript and figures

1. Throughout the manuscript, the authors use "expression levels". As protein degradation is a post-expression mechanism, "protein levels" should be used instead.

We have transitioned to using “protein levels” at many points in the manuscript.

Updated throughout the manuscript

1. Degradation dynamics (time course experiments) should be shown. The only time this is done in the current version (in Fig. 4), degradation appears to be in the same range (even a bit slower) than for EL222, which does not support the claim that the "LOVtag" acts faster than other optogenetic systems controlling protein levels.

In the revised manuscript, time course data are now shown at multiple points. These include new data in Fig. 1f and Fig. S8 that demonstrate degradation at various stages of growth. Fig. S4 also shows the dynamics of degradation when comparing to the addition of exogenously expressed ClpA. We have added text in the results section to point the reader to these data. In addition, we have made minor modifications to the text in the Introduction to avoid making claims about speed comparisons. Fig. 1f, Fig. S8, Fig. S4

Results: Design and characterization of the AsLOV2-based degradation tag, Introduction

1. "Frequency" is used incorrectly for Fig. 3. A series of 5 seconds on, 5 seconds off corresponds to a frequency of 0.1 Hz (1 illumination round / 10 s), not of 0.5 Hz. What the authors indicate as "frequency" is the fraction of illumination time. However, the (correct) frequency should be given, as this is likely the more important factor.

We have changed how we calculate frequency to use the proposed definition of one pulse per time period. We updated the values in the text and in the figure. Fig. 3c

Results: Tuning frequency response of the LOVdeg tag

1. To properly evaluate the system, several additional controls are needed:a. To test for photobleaching of mCherry by blue light illumination, untagged controls should be shown for the mCherry-based experiments. Fluorescence always seems to be lower upon illumination, except for the AsLOV2*(546) data, where it cannot be excluded that fluorescence readings are saturated. Relatedly, the raw data for OD and fluorescence should be included. Showing a Western blot against mCherry in at least one case would allow to separate the effects of photobleaching and degradation.

We appreciate the suggestion and have conducted these important controls. We now include new data demonstrating that light induction does not change fluorescence levels using an untagged mCherry control, nor does it significantly affect endpoint OD levels. Based on these results, we did not perform a Western blot because there were no effects to separate. Fig. S7

b. In Fig. 2b, light + IPTG should be shown to estimate the activity of the system at higher expression levels.

We have added these to the figure. Light + IPTG modestly increases expression compared to IPTG only, likely due to the saturating level of IPTG added, which achieves near full induction. Fig. 2b

c. In Fig. 4, EL222 alone should be shown to allow a comparison with the LOVtag. From the data presented, it looks like EL222 is both slightly faster and more efficient than the LOVtag.

We have added the EL222-only case for comparison with LOVdeg only and EL222 + LOVdeg. We note that Reviewer #3 raised a similar concern. Fig. 4b

d. The effect of the used light on bacterial viability under exponential and stationary conditions should be shown.

In this revision, we have added new data on light exposure at various points during exponential and stationary phase (Fig. 1f, Fig. S8). These OD data show that growth curves are similar for all cultures, regardless of the time light is applied during the growth phase. Additionally, we also now include ODs for the photobleaching experiments. These data also show that growth is not significantly altered under continuous light exposure. Figure 1f, Fig. S7b

1. The claim that "Post-translational control of protein function typically requires extensive protein engineering for each use case" is not correct. The authors should discuss alternative options, e.g. based on dimerization, more extensively and in a less biased manner.

We have toned down the language in this location and at other points in the manuscript. However, we maintain that other types of post-translational control, such as dimerization or LOV2 domain insertion, require more protein engineering than inserting a degradation tag. For example, we and others have directly demonstrated this in previous work (e.g. DOI:10.1021/acssynbio.9b00395, 10.1101/2023.05.26.542511, 10.1038/s41467-023-38993-6), where numerous split site or insertion variants need to be screened and fine-tuned for successful light control. In contrast, a degradation mechanism has the potential to require less fine tuning to achieve a light response. We have included the above sources to clarify this point. Introduction, Results: Modularity of the LOVdeg tag

Minor points1. In Suppl. Fig. 1, amino acid numbers seem to be off. Also, the alterations in iLID (compared to AsLOV2) that are not used in "LOVtag" appear to be missing and the iLID sequence incorrect, as a consequence.

Thank you for catching this. The number indices in Fig. S1 have been corrected. We also realized we were reporting the iLID(C530M) variant in our amino acid sequence and have reverted the 530M back to C. Fig. S1

1. Why is AsLOV2*(543) more efficiently degraded than AsLOV2(543) (blue column in Fig. 1d) when the dark state should be stabilized in AsLOV2*(543)?

We are not sure of the exact reason for the increased degradation response in the AsLOV2*(543) variant. It may be that the dark-state stabilizing mutations introduced also have more favorable interactions with degradation machinery, although this is highly speculative.

1. Why does the addition of EL222 reduce protein levels so strongly in the dark for CpFatB1* (Fig. 5)?

We believe this effect stems from the EL222 responsive promoter (PEL222). With LOVdeg only, CpFatB1* is expressed from an IPTG inducible promoter (PlacUV5) whereas EL222 responsive constructs necessitate a promoter switch containing an EL222 binding site. We have clarified this point and expanded our discussion of these results.

Results: Optogenetic control of octanoic acid production

1. Fig. 2f / S10 are difficult to interpret. Why does illumination only lead to a significant effect at 2.5 and 5 µg/ml and not at lower concentrations, where the degradation system would be expected to be most efficient?

We have expanded our discussion on these results to explain that this likely stems from basal protein levels of AcrB-LOVdeg in the light that can provide resistance at low antibiotic concentrations. We have also added new controls to this figure to show the chloramphenicol sensitivity of a ΔacrB strain and a ΔacrB strain with an IPTG-inducible version of acrB with no induction, demonstrating the lowest achievable chloramphenicol resistance from a standard inducible system.

Results: Modularity of the LOVdeg tag, Fig. 2f-g

1. Fig. 2f / S10 do not measure the MIC (which is a clearly defined value), but the sensitivity to Chloramphenicol.

We have changed the text to use the term chloramphenicol sensitivity instead of MIC. Results: Modularity of the LOVdeg tag

1. "***" in Fig. S1 should be explained.

We have removed the ‘***’ to avoid confusion. Fig. S1

1. The fold-change differences between light and dark, indicated in some selected cases, should be listed for all figures.

We have added fold-change values where appropriate. Fig 1d, Fig. 2b

**Reviewer #2:**

**Public Review:**
In this manuscript the authors present and characterize LOVtag, a modified version of the bluelight sensitive AsLOV2 protein, which functions as a light-inducible degron in *Escherichia coli*. Light has been shown to be a powerful inducer in biological systems as it is often orthogonal and can be controlled in both space and time. Many optogenetic systems target regulation of transcription, however in this manuscript the authors target protein degradation to control protein levels in bacteria. This is an important advance in bacteria, as inducible protein degradation systems in bacteria have lagged behind eukaryotic systems due to protein targeting in bacteria being primarily dependent on primary amino acid sequence and thus more difficult to engineer. In this manuscript, the authors exploit the fact that the J-alpha helix of AsLOV2, which unwinds into a disordered domain in response to blue light, contains an E-A-A amino acid sequence which is very similar to the C-terminal L-A-A sequence in the SsrA tag which is targeted by the unfoldases ClpA and ClpX. They truncate AsLOV2 to create AsLOV2(543) and combine this truncation with a mutation that stabilizes the dark state to generate AsLOV2*(543) which, when fused to the C-terminus of mCherry, confers light-induced degradation. The authors do not verify the mechanism of degradation due to LOVtag, but evidence from deletion mutants contained in the supplemental material hints that there is a ClpA dominated mechanism. They demonstrate modularity of this LOVtag by using it to degrade the LacI repressor, CRISPRa activation through degradation of MCP-SoxS, and the AcrB protein which is part of the AcrAB-TolC multidrug efflux pump. In all cases, measurement of the effect of the LOVtag is indirect as the authors measure reduction in LacI repression, reduction in CRISPRa activation, and drug resistance rather than directly measuring protein levels. Nevertheless the evidence is convincing, although seemingly less effective than in the case of mCherry degradation, although it is hard to compare due to the different endpoints being measured. The authors further modify LOVtag to contain a known photocycle mutation that slows its reversion time in the dark, so that LOVtag is more sensitive to short pulses of light which could be useful in low light conditions or for very light sensitive organisms. They also demonstrate that combining LOVtag with a blue-light transcriptional repression system (EL222) can decrease protein levels an additional 269-fold (relative to 15-fold with LOVtag alone). Finally, the authors apply LOVtag to a metabolic engineering task, namely reducing expression of octanoic acid by regulating the enzyme CpFatB1, an acyl-ACP thioesterase. The authors show that tagging CpFatB1 with LOVtag allows light induced reduction in octanoic acid titer over a 24 hour fermentation. In particular, by comparing control of CpFatB1 with EL222 transcriptional repression alone, LOVtag, or both the authors show that light-induced protein degradation is more effective than light-induced transcriptional repression. The authors suggest that this is because transcriptional repression is not effective when cells are at stationary phase (and thus there is no protein dilution due to cell division), however it is not clear from the available data that the cells were in stationary phase during light exposure. Overall, the authors have generated a modular, light-activated degron tag for use in Escherichia coli that is likely to be a useful tool in the synthetic biology and metabolic engineering toolkit.

We thank Reviewer #2 for the constructive feedback. In the updated manuscript, we now include data demonstrating degradation at different growth stages and address other points brought up in the review to improve understanding of the degradation tag.

Overall, the authors present a well written manuscript that characterizes an interesting and likely very useful tool for bacterial synthetic biology and metabolic engineering. I have a few suggestions that could improve the presentation of the material.Major Comments:• Could the authors clarify, perhaps through OD measurements, that the cultures in the octanoic acid experiment are actually in stationary phase during the relevant light induction. It isn't clear from the methods.

We have updated the Methods to clarify that the cells are entering stationary phase (OD600 = 0.6) when light is either kept on or turned off for production experiments. Production is continued for the following 24 hours. Note that we now show OD measurements in a separate set of experiments (Fig. 1f, Fig. S8).

Methods: Octanoic acid production experiment. Fig. 1f, Fig. S8

• Can the authors clarify why there is an overall decrease in protein in the clpX deletion? And is it this initial reduction that is the source of the change in fold in 1C? Similarly, for hslU is it because overall protein levels are higher with the tag? In general, I feel that the interpretation of Supplemental Figures S6-S10 could be moved in more detail to the main text, or at least the main takeaway points. But this is a personal preference, and not necessary to the major flow of the story which is about the utility of the LOVtag tool.

As shown in Fig. S5, expression of mCherry without any degradation tag is decreased in a clpX knockout strain compared to wild type. This difference may be the result of reduced cell health, and we now note this in the text. The strains shown in Fig. 1c are in wild type cells with normal expression, so this is not the source of the fold change. As for hslU, we agree it is interesting that expression seems to increase. However, the increase is modest and could stem from gene network regulation differences in that strain compared to wild type and may not be related to LOVdeg tag degradation. Each endogenous protease is involved in a wide range of functions within the cell, and it is unknown how global gene expression is impacted. We acknowledge the suggestion of moving the protease results to the main text, but we have ultimately elected to keep these data in the Supplementary Information to maintain the flow in the manuscript. However, we have added additional text pointing the reader to the Supplemental Text and include a brief summary of the findings in the main text.

Results: Design and characterization of the AsLOV2-based degradation tag

• What is the source of the poor repression in Figure 2D?

Presumably, this stems from low levels of the CRISPRa MCP-SoxS activator, even in the presence of light. We have added this point to the text.

Results: Modularity of the LOVdeg tag

• In general, it would be nice to have light-only controls for many of the experiments to validate that light is not affecting the indicated proteins or their function.

We thank the reviewer for this suggestion and note that Reviewer #1 raised a similar concern. We have now included light-only data for a strain containing IPTG-inducible mCherry without the LOVdeg tag (Fig. S7). These data show that light itself, at the levels used in this study, does not affect mCherry expression or cell growth. This strain serves as a direct control for data presented in Fig. 1 and Fig. 2b, as the systems are identical except for the addition of the LOVdeg tag onto either mCherry or the LacI repressor. Additionally, the control translates to other experiments since mCherry is used as a reporter for other systems in this study. Fig. S7

• It would be nice to directly measure the function of the tool at different phases of *E. coli* growth to show directly that protein degradation works at stationary phase, rather than the more indirect measurements used in the octanoic acid experiment.

We thank the reviewer for this suggestion, which significantly strengthens our results. We have added an experiment that tests the LOVdeg tag at different phases of growth (Fig. 1f, Fig. S8). In this experiment, cultures are growth from early exponential to stationary phase, and light is introduced at various points. Exposure windows of 4 hours, ranging from early exponential to stationary phase, all show functional light inducible degradation. Fig. 1f, Fig. S8.

Results: Design and characterization of the AsLOV2-based degradation tag

Minor Comments:• It would be nice to make clear that the data in S6d and S7 is repeated, but with the HslUV data in S7.

We clarified this point in the caption of Fig. S4 (the former Fig. S7 in the original manuscript). Fig. S4 caption

• Why was 5s picked for the frequency response in Figure 3

We picked 5s because (1) it is a substantially shorter timescale than overall degradation dynamics seen for the LOVdeg tag, and (2) we found that shorter pulses could not be reliably achieved with the light stimulation hardware and software we used (Light Plate Apparatus with Iris software). To ensure high fidelity pulses, we opted for 5 second pulses that we empirically determined to be stable throughout long experiments. We have added text clarifying this. Results: Tuning frequency response of the LOVdeg tag

**Reviewer #3:**

**Public Review:**
The authors present the mechanism, validation, and modular application of LOVtag, a light-responsive protein degradation tag that is processed by the native degradosome of *Escherichia coli*. Upon exposure to blue light, the c-terminal alpha helix unfolds, essentially marking the protein for degradation. The authors demonstrate the engineered tag is modular across multiple complex regulatory systems, which shows its potential widespread use throughout the synthetic biology field. The step-by-step rational design of identifying the protein that was most dark stabilized as well as most light-responsive for degradation, was useful in terms of understanding the key components of this system. The most compelling data shows that the engineered LOVTag can be fused to multiple proteins and achieve light-based degradation, without affecting the original function of the fused protein; however, results are not benchmarked against similar degradation tagging and optogenetic control constructs. Creating fusion proteins that do not alter either of the original functions, is often difficult to achieve, and the novelty of this should be expanded upon to drive further impact.

We appreciate the feedback from Reviewer #3 to improve the manuscript. We have included important controls and benchmarking experiments to address the reviewer’s concerns, which are detailed in the points below.

Benchmarking:The similarity between the L-A-A sequence of SsrA and the E-A-A sequence of LOVtag is one of the pieces of evidence that led the authors to their current protein design. The differences in degradation efficiency between the SsrA degradation tag and LOVtag are not shown, and benchmarking against SsrA would be a valuable way to demonstrate the utility of this construct relative to an established protein tagging tool.

We thank the reviewer for suggesting an experiment to benchmark performance. We have added new experimental data where a full length SsrA tag is added to a fusion protein of nearly identical size (mCherry-iLID), allowing us to directly compare performance to mCherryLOVdeg (Fig. S9). These results show that light inducible control with LOVdeg tag decreases protein expression levels to near those achieved with the native SsrA tag. Fig. S9.

Results: Design and characterization of the AsLOV2-based degradation tag

Additionally, there is a lack of an EL222-only control presented in Figure 4b and in the results section beginning with "Integrating the LOVtag with EL222...". Without benchmarking against this control the claim that "EL222 and the LOVtag work coherently to decrease expression" is unsubstantiated. No assumptions of synergy can be made.

We appreciate this comment and note that Reviewer #1 raised a similar concern. We have added data to Fig. 4b with an EL222-only control for comparison. Fig. 4b

The dramatic change in dark octanoic acid titer between the EL222, LOVtag and combined conditions are surprising, especially in comparison to the lack of change in the dark mCherry expression shown in Figure 4b. This data is the only to suggest that LOVtag may perform better than EL222. However, the inconsistencies in dark state regulation presented in the two experiments, and between conditions in this experiment bring the latter claim to question. A recommendation is that the authors either repeat this experiment, or comment on the observed discrepancy in dark state octanoic acid titers in their discussion.

First, a key difference between the data presented in Fig. 4 and Fig. 5 is that the production experiment is conducted over a long time period (24 hours) and the EL222/LOVdeg reporter experiment is conducted over 5 hours. Likely, performance differences between EL222 and the LOVdeg tag become more pronounced as protein accumulation occurs. Second, the LOVdeg only construct is expressed from a non-EL222 promoter which is able to achieve higher expression (see response to Reviewer #1, Minor point #3). Lastly, a convoluting factor is that the relationship between expression of CpFatB1* and octanoic acid production is not completely linear, and there are likely thresholds or expressions windows that result in similar endpoint titers. We agree a more detailed examination of how CpFatB1* changes over the course of the production period would be very interesting. However, this is beyond the scope of the present study, whose goal is to introduce and showcase the utility of the LOVdeg tag as a tool. We have added new discussion on this in the Results section to clarify some of these points. We have also repeated all experiments in Fig. 4 and consistently see the LOVdeg tag performing as well as or better than EL222. As noted in the remarks to all reviewers, these data have been updated in the revised manuscript.

Results: Optogenetic control of octanoic acid production. Fig. 4d

Based on the methodology presented, no change in the duration in light exposure was tested, even though this may be an important part of the system response. The on/off, for example in Figure 4b, is either all light or all dark, but they claim that their system is beneficial especially at stationary phase. The authors should consider showing the effects of shifting from dark to light at set intervals. (i.e. 1 hr dark then light, 2hr dark until light, etc.) This data would also aid in supporting the utility of this tag for controlling expression during different growth phases, where light may be used after the cells have reached a certain phase.

We have added new data showing the effect of light stimulation at different times in the growth cycle (see response to Reviewer #2, bullet point #5). These data demonstrate that the LOVdeg tag performs well at various points in the growth cycle. Fig. 1f, Fig. S8.

Results: Design and characterization of the AsLOV2-based degradation tag

Minor Revisions Figures:Figure 1:More clarity is needed in the naming conventions for this figure and in the body of the text. For example, a different convention than 546 and 543 should be used to refer to the full and truncated lengths of the tag. It would greatly aid understanding for this to be made more clear. The authors could simply continue to use "full" and "truncated" to refer to them. In addition, the term "stabilizing mutations" in 1c could be changed to read "dark state stabilizing mutations" to aid in clarity.

When describing the design of the LOVdeg tag, we opted towards a more technically accurate description over clarity in order to make our engineering process easily comparable to other LOV2 systems. As such, we kept the number-based nomenclature (543 or 546) to represent the domain within the phototropin 1 protein from Avena sativa (AsLOV2). The domain used in this study, and many other studies, are only amino acids 404-546, i.e. not the full sequence, thus saying simply ‘full’ or ‘truncated’ is not technically accurate. We believe the detailed nomenclature, which is limited to one section, is important to provide clarity on exactly what we used for protein engineering. In the revised version we introduce the nickname “LOVdeg” tag earlier and use it throughout the rest of the manuscript.

Results: Design and characterization of the AsLOV2-based degradation tag

1b It is not clear that this is the dark state stabilized structure in the figure, but is referred to as such only in the body of the text.

We have added text in the manuscript to clarify this is AsLOV2, not iLID, and have labeled it in the figure caption as well.

Results: Design and characterization of the AsLOV2-based degradation tag

1d. Fold change is reported in Figure 2d, and may be relevant to include those values in 1d as well.

Done. Fig. 1d

1e. It is not clear which tag is being used in this bar plot. Please specify that this is the dark state stabilized, truncated tag.

We have added a title to the plot and language to the caption, both of which clarify this point. Fig. 1e

In addition, the microscopy images provided in supplemental material should be included in the first figure as it adds a compelling observation of LOVtag activity.

We are pleased to hear that the microscopy results are beneficial, however we elected to leave them in Supplementary to preserve the flow of the manuscript in the text surrounding Fig. 1.

Figure 2:2d. It is unclear what the 2.5x fold change is relative to (the baseline or the dark)

We have added a line in the figure to clarify the comparison being made. Fig. 2d

2f. More discussion can be added to describe what concentration of chloramphenicol is biologically/bioreactor relevant.

Our previous studies on the relationship between AcrAB expression and mutation rate (cited in the text) were carried out at a concentration within the range in which the LOVdeg tag is effective (5 μg/ml), suggesting this range to be relevant to tolerance and resistance.

Figure 3:We recommend that this data and discussion are better suited for supplementary figures. The results shown here essentially recapitulate the same findings of Zoltowski et al., 2009. In addition, the paper describing this mutation should be cited in this figure caption in addition to the body of the text

Although these results are in line with previous findings, we believe this dataset is important for several reasons. First, the agreement with known mutations validates the unfolding-based mechanism for degradation control. Second, degradation that is contingent on unfolding of LOV2 offers a direct actuating mechanism of photocycle properties. Other systems, like that in Zoltowski et al., examine properties of purified proteins but lack the mechanism to translate its effect in live cells. This figure demonstrates how degradation can do so and lays the groundwork for degradation-based frequency processing circuits. Last, there are discrepancies between photocycle kinetics in situ, as reported by Li et al. (DOI: 10.1038/s41467-020-18816-8), and in cell-free studies such as in Zoltowski et al. The studies use different methods of measuring photocycle kinetics (in situ vs cell-free). This dataset substantiates relaxation times from Li et al. and suggests cell-free relaxation time constants are over estimated relative to our live cell results.

Figure 4:There is a lack of an EL222-only control presented in Figure 4b. Without this data present, the claim that "EL222 and the LOVtag work coherently to decrease expression" is unsubstantiated. No assumptions of synergy can be made.

We have added EL222-only data to the figure; we note that Reviewer #1 made a similar request. Figure 4b

ManuscriptResultsDesign and characterization...Due to the extensive discussion of ClpX at the beginning of this section, more of the results on evaluating the binding partners and mechanism of LOVtag degradation should be presented in the main body of the manuscript and not in supplementary materials.

To maintain flow of the manuscript and focus on how the LOVdeg tag works as a synthetic biology tool, we have opted to keep this section in the Supplement Information, but have several lines in the text related to Fig. 1 that point the reader to this material. Results: Design and characterization of the AsLOV2-based degradation tag

In the second paragraph of this section, the authors theorize that the C-terminal truncated E-AA sequence will "remain caged as part of the folded helix". How did the authors determine this? Was there any evidence to suggest that the truncated state would be any more responsive than the full length sequence? More data or rationale may need to be introduced to support the overall hypothesis presented in this paragraph.

We determined this by examining the crystal structure which shows that the E-A-A sequence is part of the folded helix. As seen in Fig. 1b, addition of amino acids after the EAAKEL sequence would not be part of the folded helix which ends prior to the terminal leucine. We added text to clarify our logic.

Results: Design and characterization of the AsLOV2-based degradation tag

The similarity between the L-A-A sequence of SsrA and the E-A-A sequence of LOVtag is one of the pieces of evidence that brought the authors to their current protein design. The differences in degradation efficiency between the SsrA degradation tag and LOVtag are not clear, and benchmarking against SsrA would be a valuable way to demonstrate the utility of this construct relative to an established protein tagging tool.

We added an SsrA comparison to benchmark the system. Fig. S9

Results: Design and characterization of the AsLOV2-based degradation tag

Tuning frequency and response...Overall the results presented in this section essentially recapitulate the effects that mutation presented in Zoltowski et. al., 2009 have on AsLOV2 dark state recovery and although this is a useful observation of LOVtag performance, a recommendation is to move this into a supplementary section.

See above response to Fig. 3 comment.

Integrating the LOVtag with EL222...The claim is made in this section that LOVtag and EL222 work synergistically, however the experiments presented do not test repression due to EL222 activity alone. Without benchmarking against this control, the claim of synergy is not supported and we recommend that the authors perform this experiment again with the EL222-only control.

We have added this important control. Fig. 4b

DiscussionThe statement "the LOVtag can easily be integrated with existing optogenetic systems to enhance their function" is not substantiated without benchmarking LOVtag against an EL222- only control. As mentioned above this condition should be included in the experiments discussed in Figure 4 and in the section "Integrating the LOVtag with EL222.."

We added EL222-only regulation to benchmark the LOVdeg tag and LOVdeg + EL222 experiments. Fig. 4b

ExperimentsApplications:The application of this tag to the metabolic control of octanoic acid production could be more impactful. For instance, using the LOVtag with two different enzymes to change the composition of long/short chain fatty acids with light induction., Or possibly integrating the tag into a switch to activate production. However, the authors address that "decreasing titers is not the overall goal in metabolic engineering" in their discussion, and therefore the pursuit of this additional experiment is up to the authors' discretion.

We appreciate the suggestions for further applications of the LOVdeg tag. We envision that follow up studies will focus on the application of the LOVdeg tag in metabolic engineering. However, this will require significant development of production systems. We believe this to be out of the scope of this work, where the goal is to present the design and function of the LOVdeg tag as a tool.